# Semantic Mechanical Search with Large Vision and Language Models

**Satvik Sharma**[1*]**, Huang Huang**[1*]**, Kaushik Shivakumar**[1]
**Lawrence Yunliang Chen**[1]**, Ryan Hoque**[1]**, Brian Ichter**[2]**, Ken Goldberg**[1]

**Abstract:** Moving objects to find a fully-occluded target object, known as *mechanical search*, is a challenging problem in robotics. As objects are often organized semantically, we conjecture that semantic information about object relationships can facilitate mechanical search and reduce search time. Large pretrained vision and language models (VLMs and LLMs) have shown promise in generalizing to uncommon objects and previously unseen real-world environments. In this work, we propose a novel framework called Semantic Mechanical Search (SMS). SMS conducts scene understanding and generates a semantic occupancy distribution explicitly using LLMs. Compared to methods that rely on visual similarities offered by CLIP embeddings, SMS leverages the deep reasoning capabilities of LLMs. Unlike prior work that uses VLMs and LLMs as end-to-end planners, which may not integrate well with specialized geometric planners, SMS can serve as a plug-in semantic module for downstream manipulation or navigation policies. For mechanical search in closed-world settings such as shelves, we compare with a geometric-based planner and show that SMS improves mechanical search performance by 24% across the pharmacy, kitchen, and office domains in simulation and 47.1% in physical experiments. For open-world real environments, SMS can produce better semantic distributions compared to CLIP-based methods, with the potential to be integrated with downstream navigation policies to improve object navigation tasks. Code, data, videos, and Appendix are available here.

**Keywords:** Vision and Language Models, Mechanical Search, Object Search

## 1 Introduction

Mechanical search, where a robot manipulates objects and/or navigates to find a fully occluded target object [1, 2], is a challenging robotics problem. Prior work has shown success in revealing the desired object by manipulating the occluding objects [3, 4, 5], obtaining new observations after rotating the camera [6], or navigating to new locations [7, 8]. However, generalization to unseen environments remains challenging due to the numerous long-tail objects present in the real world.

Environments are often organized semantically, for example, toothpaste is often stored in a home bathroom near toothbrushes. In this paper, we explore how LLMs can provide such semantic relationships to facilitate mechanical search. Large vision and language models (VLMs and LLMs) show promise for such relationships as they are pretrained on internet-scale data which empirically captures knowledge of semantics. A large body of prior work has shown that these models can provide good visual representations [9, 10, 11, 12, 13], ground language instructions [14, 15, 16, 17, 18, 19, 20], and serve as planners out of the box [21, 22, 23, 24, 25, 26]. CLIP [27] is a commonly-used interface to associate vision and language, and many works [7, 8, 28, 29] use it to build semantic scene representations and show improved performance on object query and navigation tasks. However, while informative, the dot product of CLIP text and image embeddings lacks deep reasoning

---

[*]Equal Contribution
[1]AUTOLab at the University of California, Berkeley
[2]Google Brain

7th Conference on Robot Learning (CoRL 2023), Atlanta, USA.

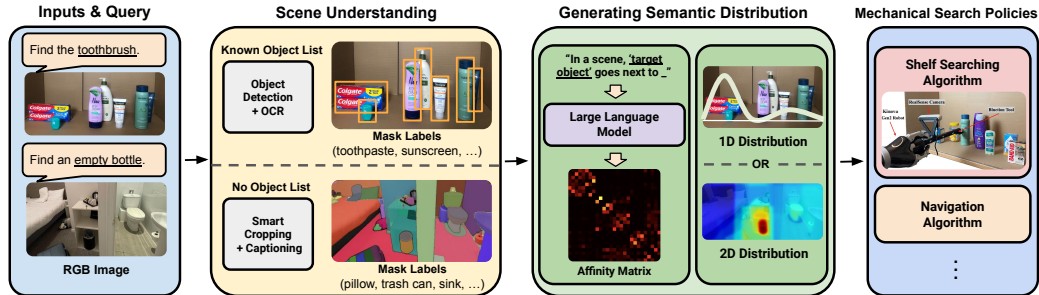

Figure 1: **Overview of Semantic Mechanical Search (SMS).** SMS accepts as input a scene image and a desired target object. It applies an object detection, or segmentation algorithm combined with captioning as necessary when object lists are unavailable. SMS then uses an LLM to compute affinities between detected objects to the target object, and it uses these affinities to output a semantic occupancy distribution which can be used for downstream mechanical search policies.

capabilities and sometimes behaves as a bag-of-words [30]. As such, CLIP is most useful for localizing objects that are already visible somewhere in the scene or map [13, 8], a property that many open-vocabulary object detectors build on [31, 32, 33]. When the target object is fully occluded, CLIP alone may not provide enough clues about potential target object locations.

LLMs demonstrate advanced reasoning and planning capabilities [34]. Many prior works [25, 35, 7] use VLMs and LLMs as end-to-end planners for both perception and planning. While such paradigms benefit from the semantic reasoning abilities of LLMs, they do not handle additional information that cannot be easily expressed through language and may not integrate well with other domain-specific policies. For example, for mechanical search on shelves, the geometric properties of objects provide valuable cues for identifying potential target object positions, and various algorithms have been proposed for handling uncertainty and planning ahead [3, 4, 5]. Likewise, for object navigation, prior research has explored exploration and navigation strategies that are independent of semantic understanding [36, 37, 38, 39]. As such, decoupling semantic reasoning and geometric planning may allow flexible integration with task-specific modules for various downstream settings.

We propose Semantic Mechanical Search (SMS), which generates an explicit intermediate representation, a *semantic occupancy distribution*, as a plug-in semantic module for existing mechanical search algorithms. This distinguishes it from prior work where VLMs and LLMs serve as end-to-end planners doing both semantic reasoning and action planning. With the goal of adding semantic reasoning to existing search policies, we study two questions: (1) Can a semantic distribution facilitate mechanical search? (2) What is the best way to generate this semantic distribution? For the second question, we hypothesize that translating image features into language features (with VLMs) first and then extracting semantic distributions from only language features (with LLMs) can outperform VLM-only methods that most current works use. We show that, rather than burdening VLMs (e.g. CLIP) with both object detection and reasoning, decoupling these two tasks leads to better results as the LLM language feature space is better at capturing semantic relations. For the first question, we show SMS can be easily integrated with a geometric shelf searching algorithm [3] to improve performance for closed-world environments such as pharmacy shelves with known object lists. In closed-world settings where object lists are available, SMS uses an open vocabulary object detection model [40] refined with Optical Character Recognition (OCR) to identify objects. In open-world settings where object lists are unavailable, SMS combines segmentation [41] and image captioning [42] to generate object mask descriptions.

This paper makes three contributions:

1. SMS, a novel framework that uses pretrained VLMs and LLMs to synthesize semantic occupancy distributions that can be easily integrated to enhance mechanical search policies;

2. A way of using LLMs to augment the reasoning capabilities of VLMs for generating better semantic distributions, with evaluations of semantic distribution quality in both closed-world and open-world settings.

3. Closed-world experiments for mechanical search on shelves showing that SMS improves a geometric-based planner by 24% across the pharmacy, kitchen, and office domains in simulation and 47% in real, and a preliminary study of SMS in open-world settings.

## 2 Related Work and Preliminaries

### 2.1 Mechanical Search

Mechanical search [1, 2] refers to a broad class of robotics problems on searching for occluded and out-of-view objects via manipulation and navigation. In the former case, bin [1] and shelf environments [43, 44, 45, 46, 47] are widely studied, where intelligent estimation and manipulation planning based on possible locations of the hidden target object significantly affects the search efficiency. Many prior work uses only geometric priors [1, 4, 5, 3, 48]. A number of authors have also explored using semantic context object information [49]. Kollar and Roy [50] obtain co-occurrence statistics from web-based ontologies and Wong et al. [51] extend the approaches to occluded target objects. Kurenkov et al. [2] propose a hierarchical model to integrate semantic and geometric information and learn in simulation. However, they manually craft semantic categories, which are also sparse and can not accurately and scalably reflect real-world distributions. Instead, we harness large pretrained models to extract open-vocabulary semantic information zero-shot.

There are many types of navigation tasks, such as point goals [52, 53, 54], image goals [55, 56], and object goals [57, 58]. Finding out-of-view objects is an object goal navigation task, and the problem is also known as active visual search [59, 60]. Classical geometry-based methods typically first build a map [61, 62] and then perform planning [63, 64]. Learning-based methods typically use reinforcement learning trained in simulation [65, 57, 66, 53, 67, 68, 69, 70, 71], through YouTube videos [58], or by querying the Internet [72] to learn semantics and efficient exploration strategies. Recently, many works have explored using LLMs and VLMs out-of-the-box for semantic scene understanding [73] and zero-shot object navigation, which this work belongs to. The most common strategy is to use CLIP features [27] obtained from pretrained open-vocab detectors [74, 75] as in VL-Maps [7] and OpenScene [76] or from region proposals models as in CLIP-Fields [77], ConceptFusion [78], and NLMaps-SayCan [79] and fuse them into 3D point clouds or implicit representations [30]. The constructed representations can then be used for open-vocabulary target queries to locate the object and perform navigation. Gadre et al. [8] propose a family of methods to adapt CLIP and open-vocabulary models to localize target objects. Through a systematic comparison, they find OWL-ViT detector [80] works best, followed by patchifying images to obtain separate CLIP embeddings and compute similarity with text embeddings. In this work, instead of using the similarity of CLIP embeddings to construct relevancy maps [81], we use the LLM feature space to explicitly reason about the object's semantic relationships and show that SMS outperforms these two methods.

### 2.2 Natural Language in Robotics

Grounding natural language instructions is a widely-studied problem in robot navigation [82, 83, 84, 85, 86, 87, 88, 89, 90, 91, 92, 93], human-robot interaction [94, 95], and is increasingly studied in the manipulation literature [96, 97, 98, 99]. While classical methods commonly rely on semantic parsing and factor graphs [87, 91, 94], end-to-end learning and leveraging pretrained models are now the most popular paradigms thanks to advances in deep learning and LLMs. Examples include language-conditioned imitation learning [100, 19, 16, 18, 20, 101], language-conditioned reinforcement learning [102, 103, 14], and online correction of robot policies through language feedback [104, 105]. In particular, pretrained image encoders and open-vocabulary object detectors have enabled generalization to novel object queries at test time [25, 24, 13]. In this work, we also take in novel object targets specified using natural language, but since the target objects are not visible in the scene, the robot instead needs to detect and localize other objects and reason about their relationships. This is particularly challenging in an open-world environment when the set of possible objects is unknown, making object detectors significantly less accurate. Our method shares similarity with HOLM [6], which uses an LLM to hallucinate nearby objects in partially observable scenes based on semantics computed from affinity scores. However, it relies on an object list and only considers camera adjustment actions in simulation. We relax this assumption of accessing object

lists [7, 76, 6], propose a pipeline for generating object labels without access to any object lists, and generate semantic distributions for open-world environments.

Many studies have also used LLMs as a planner by letting them break down tasks through step-by-step reasoning [22, 23, 24, 25, 35, 106] or directly write code [21, 26]. While these end-to-end planning paradigms benefit from the deep reasoning abilities of LLMs, it's not straightforward to incorporate additional non-language information and integration with domain-specific policies. The latter is particularly valuable when the task is more complex and a flexible generalist LLM can benefit from specialized searching and planning algorithms developed by the robotics community. We propose decoupling semantic reasoning and geometric planning; rather than directly output primitive instructions from image observations, SMS uses LLMs' semantic reasoning from its feature space into a semantic distribution that specialized planning and manipulation policies can use.

### 2.3 Occupancy Distribution

An occupancy distribution indicates the probability of each pixel in the image containing the target object's amodal segmentation mask [1]. Prior works [1, 3, 4, 5] have utilized geometric information to generate spatial occupancy distributions by considering object geometries and camera perspective (e.g., tall target objects cannot be occluded by short objects and objects in the center of an image occlude more areas) to facilitate the search. Huang et al. [4] propose the LAX-RAY system, which uses a neural network to predict the spatial occupancy distribution. A greedy policy called Distribution Area Reduction (DAR) uses this distribution to greedily reduce the overlap between objects and the distribution the most. SMS generates the occupancy distribution using semantic information, which can then be combined with the LAX-RAY spatial distribution for downstream search.

## 3 Problem Statement

We consider a partially observable environment that contains a target $\mathcal{O}_T$ and $N$ other objects $\{\mathcal{O}_1, ..., \mathcal{O}_N\}$. We assume the scenes in the environment are semantically organized, meaning that the starting state of the environment is sampled proportionally to their approximate likelihood of occurrence in the real world. With this assumption of semantically organized scenes, the target object location probability is proportional to object pair affinities. States $s_t \in \mathcal{S}$ consist of the full geometries, poses, and names of the objects in the scene at timestep $t$, and observations $y_t \in \mathcal{Y} = \mathcal{R}^{H \times W \times 3}$ are RGB images from a robot-mounted RGB camera at timestep $t$. Given the name of the target object and the observation $y_t$, the goal is to generate a useful dense occupancy distribution that encodes semantic affinities (with respect to the target object).

## 4 Semantic Mechanical Search

We propose SMS, a framework using VLMs and LLMs to create a dense semantic distribution between a scene and the target object to be used for downstream tasks. Fig. 1 visualizes the pipeline. SMS first uses VLMs to perform scene understanding by creating mask-label pairs to densely describe all image portions. It then uses an LLM to generate affinity scores between the labels and the target object. We spatially ground these affinities using the labels' corresponding masks. In this way, we densely represent the affinities between a target object and all parts of a scene using an LLM. SMS can be applied to two common situations: 1) a closed world where all objects in the scene are a subset of a known list and 2) an open world where some objects in the scene are previously unseen.

### 4.1 Scene Understanding

The goal of scene understanding is to generate mask-label pairs that characterize the scene.

**Object Detection + OCR** When an object list is available, we use an open vocabulary object detection model, specifically ViLD [40], to obtain object segmentation masks and labels from an RGB image. We also find using Keras Optical Character Recognition (OCR) [107] can improve the quality of the ViLD object detection, by increasing the mean average precision (mAP) on 100 pharmacy scene images from 2.4 to 28.9 (Table 5). Details are in the Appendix.

**Crop Generation + Image Captioning** When an object list is not available, many open vocabulary detectors such as ViLD cannot be used. We instead create image crops and use a VLM for crop

captioning, specifically BLIP-2 [42], to convert object crops to their text descriptions. We ask for the dominant objects in each crop for less noisy captions. We generate crops that are both object-centric (using Segment-Anything (SAM) [41]) for better object boundaries in the semantic distribution and general multiscale, overlapping crops that help encode large-scale semantic information.

## 4.2  Creating the Semantic Distribution

We consider two ways to use a language model to generate affinity scores for the semantic distribution. **(1) SMS-LLM:** We iterate over all the mask-label pairs and query the LLM with a specific prompt: "*I see the following in a room:* {*label*}. *This is likely to be the closest object to* {*target object*}". This prompt directly represents the probability of the target object given we see the label. Since object labels are contained within the prompt, we do not need to normalize to account for the prior. A similar prompt with the label and the target object switched would also provide affinity scores between objects but would then have to be normalized to account for that object's prior. The affinity score for the target object with each label is the completion probability for the tokens that represent the target object. We generate a semantic distribution from these affinity scores and detected objects. The semantic distribution models the probability of the target object occupying each location, which we approximate to be proportional to the affinity score between the target and the object closest to that location. To account for noise, we apply spatial smoothing using a Gaussian kernel with std $\sigma$. More details are in the Appendix. **(2) SMS-E:** An alternative method we explore is to use a language embedding model (e.g. OpenAI Embedding Model [108]) to get embeddings for all labels and the target object, and obtain an affinity score between each label and the target object through the dot product between these vectors.

When there are no object lists, the Crop Generation + Image Captioning pipeline described in Section 4.1 can contain many incorrect or hallucinated labels, making the distribution noisy. To mitigate this, we use CLIP to verify the captions and not for any semantic reasoning. Specifically, we compute the CLIP dot products between the image crops and the generated labels and weight the affinity scores by these relevance scores. To produce the final semantic distribution, each pixel receives the average of the weighted affinity scores of all the masks it belongs to. We find that averaging across multiple overlapping masks also helps reduce noises in the absence of object lists.

## 4.3  Combining with Mechanical Search Policies

**Closed-World Environments** We consider semantically organized shelves with objects from a known list. For mechanical search on shelves, the robot needs to manipulate objects in the shelves to reveal the occluded target object using pushing and pick-and-place actions. The goal is to minimize the number of actions taken to reveal the target object. Additional details are in the appendix. We use SMS as a plug-in semantic module for an existing search algorithm, LAX-RAY [4], by multiplying the semantic occupancy distribution with a learned spatial distribution that LAX-RAY generates based on geometry. We then use the DAR policy [4] to perform mechanical search. Since the search in cluttered environments requires manipulating other objects, once the search begins, the shelf may become semantically disorganized. As such, at each step in a rollout, SMS computes the semantic distribution using the object locations where each object was first discovered.

**Open-World Environments** We consider large room spaces, with semantic diversity (rooms of an office, home, aisles in a grocery store, etc.). We do not perform any manipulation in this setting and explore a downstream heuristic navigation policy that terminates when the object is within view. Given a starting position, the policy moves a fixed distance towards the highest affinity region in the image. Afterward, it takes four new images by rotating in place. We first select the desired view direction amongst the four by choosing the one that has the highest 90-percentile affinity score to ensure we are more robust to outlier affinities that may result from not having an object list. Then, after selecting the view, we again select the highest affinity point and move to that location.

## 5  Experiments

We investigate two questions: with a given downstream search policy, (1) can a semantic distribution improve search performance? and (2) what is the best way to generate a semantic distribution?

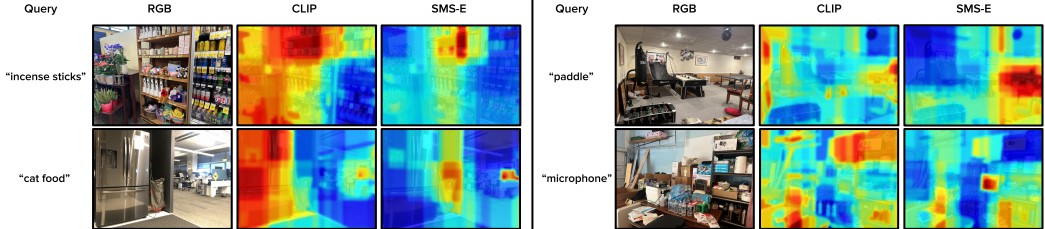

Figure 2: **Generating semantic distributions in open-world environments.** Four examples from the evaluation dataset with the 2D probability distributions generated for SMS-E and CLIP. These heatmaps are red for high-probability regions of finding the target object and blue for low probability. **Top Left:** An example of a grocery store, where the target object is "incense sticks." CLIP highlights both near the candles and near the flowers as they are somewhat visually similar to sticks, while SMS-E only highlights the candles. **Bottom Left:** An example of an office kitchen, where the target object is "cat food." CLIP gets distracted by the refrigerator and only slightly highlights the cat sign. **Top Right:** An example of a house, where the target object is "paddle." CLIP incorrectly highlights the wooden panels along the walls, while SMS-E highlights the ping pong table. **Bottom Right:** For the target word "microphone," SMS-E highlights the box with the speaker but CLIP struggles as the objects are not visually similar.

## 5.1 Evaluation of Semantic Distributions Quality

We investigate the second question first to obtain a semantic distribution for the downstream policy. We evaluate semantic distribution generation both in closed-world and open-world environments. In close-world environments, we evaluate the affinity matrix quality where the semantic distribution is generated, for the given object list. In open-world environments, we evaluate the semantic distribution quality on a dataset of real-life scenes.

### 5.1.1 Closed-World Environments

**Experimental Setup:** With an object list, the semantic distributions are directly generated from semantic affinity matrices, with rows and columns as objects from the list and the entries representing the affinities scores between objects calculated in Section 4.2. We use an object list of 27 objects in the pharmacy domain (list in Appendix). We directly compare the affinity matrix quality. We approximate a ground truth affinity matrix with Google Taxonomy as in Section B.1. The Google taxonomy provides semantic information for evaluation purposes to avoid human bias. Since it has limited categories, it cannot be used directly for objects that do not appear in the taxonomy.

**Results:** We compare the quality of the affinity matrix for the given object list generated by SMS and a CLIP-based baseline proposed by CoW [8], which uses the dot products of CLIP text embedding as the affinity scores. We compute the reduction of Jensen-Shannon Distance (JSD) [109] between the generated affinity matrix and the ground truth affinity matrix compared to the JSD between a uniform matrix and the ground truth. This quantifies the benefit SMS provides over a uniform distribution. From Table 1, we see that SMS significantly outperforms the CLIP-based method, while the SMS-LLM variant slightly outperforms the SMS-E. This suggests the reasoning capability of LLM models is more valuable for capturing semantics than CLIP embeddings. We compare other methods in Table 6 in the Appendix.

| Metric | CLIP | SMS-E | SMS-LLM |
|---|---|---|---|
| $\Delta\%$ JSD $\uparrow$ | 20.0% | 33.8% | 44.6% |

Table 1: **Closed-world semantics evaluation.** Percentage improvement of the generated semantic distributions in the pharmacy domain compared to a uniform prior, measured based on the Jensen-Shannon Distance (JSD) from the ground truth distribution.

| Method | OWL-ViT | CLIP | SMS-LLM | SMS-E |
|---|---|---|---|---|
| IoU | $0.138 \pm 0.031$ | $0.221 \pm 0.034$ | $0.345 \pm 0.039$ | $\mathbf{0.391} \pm 0.039$ |

Table 2: **Open-world semantics evaluation.** IoU results for different methods. The object detector-based method, OWL-ViT, performs poorly because even though the target objects are semantically related, many have very little visual similarity. CLIP performs worse than SMS because SMS is getting semantic similarity in a language-only latent space which can capture more nuance than the visual-language embedding space.

### 5.1.2 Open-World Environments

**Experimental Setup:** For the open-world environment, we evaluate the semantic distribution generation on a static image dataset consisting of 30 real scenes taken from 4 houses, 4 office buildings, and 3 local grocery stores. We sampled 90 objects across the three domains and chose those scenes

based on our accessibility to those places. In all scenes, the objects' numbers and placements are set by their management. All scenes and the target object list are in Appendix E. Since these scenes are large, we are interested in quantifying the accuracy of the semantic distribution along both the $x$- and $y$-axes. We annotate the ground truth search area based on the real scene and use Intersection over Union (IoU) to quantitatively evaluate the accuracy of each method.

**Results:** We evaluate the following VLM-only baselines: CLIP and OWL-ViT, the two best-performing methods found by Gadre et al. [8]. For CLIP, it uses the same crop-label pairs as SMS to generate a semantic distribution as described in Section 5.1 but with further augmentations (jittering and horizontal flipping) on those crops for better performance [73]. We threshold this distribution and create a mask to calculate IoU with the ground truth. OWL-ViT gives bounding boxes for its labels and we directly use them to calculate the IoU. We find that OWL-ViT performs better if the best bounding box is selected rather than weighting all bounding boxes by their score and thresholding that distribution. Table 2 shows the results. SMS generates semantic distributions within 35 to 45 seconds. We see that SMS outperforms the VLM-only methods including both CLIP and OWL-ViT. More examples are in the Appendix. We hypothesize that this is because CLIP focuses more on the visual appearance of the objects rather than semantic relations. This would be less of a problem for searching visible objects but is not ideal for searching objects that are outside the field of view or occluded. For example, CLIP would associate incense sticks with sticks used for gardening while LLMs would associate the incense sticks with the candles. In addition, CLIP has a "bag of words" behaviour [30], causing it to incorrectly relate "cat food" with a fridge instead of a cat sign. In contrast, LLMs have better semantic reasoning as shown in Figure 8, where "cat food" highlights the cat sign as the highest region but also highlights the gray bag because cat food could be occluded inside of a bag. Since LLMs are trained on large corpora of human language, we hypothesize that they effectively encode the semantics of both common and rare objects and are also capable of semantic reasoning (e.g. cat food can be inside the bag) beyond just creating class categories and thus are better suited for searching fully-occluded objects. SMS-E slightly outperforms SMS-LLM as they are both bottlenecked by the quality of labels from BLIP-2.

We also conduct an ablation study for each module of SMS on semantic distribution quality with results in Table 11 in the Appendix, indicating the effectiveness of cropping with SAM and CLIP weighting, and the impact of image captioning model choice.

## 5.2 Semantic Distribution Effect on Mechanical Search Performance

Given a semantic distribution, we investigate the first question by conducting simulation and real experiments in close-world environments to evaluate search performance improvement brought by the semantic distribution. We combine the semantic distribution with an existing mechanical search policy LAX-RAY as in Section 4.3.

**Experimental Setup:** For simulation, we consider a pharmacy, a kitchen, and an office domain. In addition to the 27 objects for the pharmacy domain, we consider 24 and 40 representative objects for the kitchen and office domains from the Google Product Taxonomy [110]. We generate semantically organized scenes with the procedure detailed in Appendix D.1 and examples in Figure 5. The simulation and real experiments take place within a $0.8\,\text{m} \times 0.35\,\text{m} \times 0.57\,\text{m}$ shelf environment.

### 5.2.1 Simulation Experiments

We run extensive experiments using the First Order Shelf Simulator (FOSS) from [3]. In simulation experiments, we assume perfect object detection but consider geometry for occlusion. For each domain, we generate semantically organized scenes (details in Appendix D.1) with various numbers of objects $N$=12, 15, 18, 21 with 200 scenes for each. Termination occurs when the target object becomes visible or reaches maximum action number $2N$.

For each scene, we evaluate whether SMS improves the performance of LAX-RAY [3], which only uses geometric models. We consider both SMS-E and SMS-LLM for augmenting the geometric distribution from LAX-RAY. We report two metrics: **Success rates:** The ratio of trials where the target object is found within the maximum action limit to the total number of trials. **Number of actions:** The mean and standard error of the number of actions required to reveal the target object.

| | Pharmacy Domain | | | Kitchen Domain | | | Office Domain | | |
|---|---|---|---|---|---|---|---|---|---|
| | Successes | # Actions | Δ% | Successes | # Actions | Δ% | Successes | # Actions | Δ% |
| LAX-RAY | 576/741 | 5.56 ± 0.20 | N/A | 703/770 | 3.32 ± 0.14 | N/A | 575/753 | 4.14 ± 0.19 | N/A |
| SMS-E | 591/741 | 4.18 ± 0.17 | 24.8 | **725/770** | 2.43 ± 0.10 | 26.8 | 580/753 | 4.10 ± 0.18 | 0.9 |
| SMS-LLM | **606/741** | **3.76 ± 0.14** | **32.4** | 710/770 | **2.42 ± 0.10** | **27.1** | **598/753** | **3.63 ± 0.16** | **12.3** |

Table 3: Simulation experiment results for three domains averaged over 12, 15, 18, 21 number of objects, also reported with Δ%, the percentage reduction in the number of actions compared to LAX-RAY.

We report results for all numbers of objects $N$ in the Appendix and the results averaged across all values of $N$ in Table 3. In all domains, SMS-LLM and SMS-E improve LAX-RAY performances with higher success rates and fewer search actions. In the pharmacy and office domain, SMS-LLM outperforms SMS-E, while in the kitchen domain, they perform comparably. For the office experiments, the performance improvement is relatively small. We hypothesize that this is due to a majority of the office environment consisting of generic office supplies that do not have a clear semantic categorization, making semantic prior less effective. Overall, the results suggest that SMS-LLM can serve as a semantic plug-in module and improve LAX-RAY performance in semantically arranged environments by 32.4%, 27.1%, and 12.3% in the pharmacy, kitchen, and office domains respectively while improving success rates. SMS-LLM outperforms SMS-E, indicating the quality of the affinity matrix is directly correlated with the task performance.

We also show a strong positive correlation between object detection accuracy and task performance with results and details in Appendix D.3, indicating the benefits of SMS using OCR. In addition, we show SMS are effective on different downstream policies by using SMS as the plug-in module for Distribution Entropy Reduction (DER) from [3]. The details are in Appendix D.3.1.

### 5.2.2 Physical Experiments

| Method | # Actions | Δ% | Method | # Actions | Δ% |
|---|---|---|---|---|---|
| LAX-RAY | 4.25 ± 0.64 | N/A | SMS-LLM | 2.25 ± 0.46 | 47.1 |

Table 4: Physical experiment results (12 trials each). We report the average number of actions taken to reveal the target object as well as the percentage reduction in the number of actions over the spatial neural network.

We conduct experiments on a physical pharmacy shelf. We use the Kinova Gen2 robot with a 3D-printed blade and suction tool [4] (see Figure 1). An Intel RealSense depth camera mounted on the tool provides RGBD observations. We use 3 scenes each of $N = 7, 8, 9,$ and 10 objects for a total of 12 scenes and a threshold visibility of 50% for determining success. More details are in the Appendix. As simulation results from Table 3 suggest SMS-LLM outperforms SMS-E, we evaluate SMS-LLM in physical experiments. An identical set of 12 semantically arranged scenes (starting configurations) is used for each method.

Results are shown in Table 4. We observe that SMS significantly accelerates mechanical search on shelves, reducing the average number of actions by 47.1%. In physical experiments, the noises in the depth images result in worse spatial distribution than in simulation, making the semantic distribution more critical in identifying where a target object may lie. We also conduct a preliminary navigation experiment in open-world environments as in Section 4.3 with details in Appendix D.5.

## 6 Limitations and Future Work

We present Semantic Mechanical Search (SMS), an algorithm for semantic distribution generation for a fully-occluded target object. SMS facilitates mechanical search in closed-world environments and improves semantic distribution quality for open-world environments. SMS has several limitations, which open up possibilities for future work: (1) We only evaluate one open-world navigation task with a heuristic navigation policy without large-scale evaluations of other open-world environments; (2) While SMS, which operates in the LLM feature space, generates better semantic distributions than CLIP-based method, we have not compared to other VLMs such as GPT-4 [111] or LLaVa [112] due to their inaccessibility to obtain affinity scores. VLMs with strong reasoning abilities, such as GPT-4V [113], have the potential to directly generate high-quality semantic distributions. Further applying GPT-4V in object search would be an interesting future direction. (3) SMS for closed-world relies on creating an offline affinity matrix which can take a few minutes with large object lists, while SMS for open-world takes 35 to 45 seconds for each semantic distribution (4) The performance of SMS is sensitive to the quality of each module in the framework.

## Acknowledgments

This research was performed at the AUTOLAB at UC Berkeley in affiliation with the Berkeley AI Research (BAIR) Lab. The authors are supported in part by donations from Toyota Research Institute, Bosch, Google, Siemens, and Autodesk. L. Y. Chen is supported by the National Science Foundation Graduate Research Fellowship Program under Grant No. DGE 2146752. Any opinions, findings, conclusions, or recommendations expressed in this material are those of the authors and do not necessarily reflect the views of the sponsors. We thank our colleagues who helped with the project and provided helpful feedback and suggestions, in particular Chung Min Kim and Alishba Imran.

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

# Appendix

# A   Scene Understanding

## A.1   Object detection + OCR

Because ViLD is a general-purpose detector, it cannot easily distinguish between objects belonging to the same domain (e.g., Advil versus Ibuprofen). Because of this, we use OCR with Keras OCR[107] to improve the quality of the object detections. While OCR has been used in prior work to aid object detection [114], we use text embedding combined with OCR for better performance. For each object, we concatenate the text observed on it and compute the text embedding using OpenAI Embeddings. We compute the dot product between the embeddings of the concatenated text and every class label. We normalize this probability vector by subtracting the minimum value and then adjusting the vector with some temperature. We finally multiply this by the object detection probability vector.

Let $C_i$ denote the class label of object $\mathcal{O}_i$ (e.g., "Tylenol" as opposed to the broader category "medication"); $I_i$ represent the general shape, size, and color-related features of $\mathcal{O}_i$; and $T_i$ be the detected text on $\mathcal{O}_i$. Recall that all objects belong to some class $C_i$. We calculate

$$
\begin{aligned}
&P(C_i|\, I_i, T_i) \\
&= \frac{P(I_i, T_i|C_i) \cdot P(C_i)}{P(I_i, T_i)} \\
&= \frac{P(T_i|I_i, C_i) \cdot P(I_i|C_i) \cdot P(C_i)}{P(I_i, T_i)} \\
&= \frac{P(T_i|C_i) \cdot P(I_i|C_i) \cdot P(C_i)}{P(I_i, T_i)} \\
&= \frac{P(C_i|T_i)P(T_i)}{P(C_i)} \cdot \frac{P(C_i|I_i)P(I_i)}{P(C_i)} \cdot \frac{P(C_i)}{P(I_i, T_i)} \\
&\propto P(C_i|T_i)P(C_i|I_i)
\end{aligned}
$$

as $T_i$ is independent of $I_i$ when conditioned on $C_i$, and $P(C_i)$ is uniform. This illustrates that the multiplication of the OCR probabilities and the object detection probabilities can give us a refined estimate of the category probabilities.

We test object detection performance on scenes generated through isolated perception experiments. We take RGB images of 100 scenes of the Pharmacy domain using a high-resolution camera and study the effect of having OCR. Results for this experiment are in Table 5. As is standard in the computer vision literature, we report mAP (mean Average Precision) averaged over intersection-over-union (IOU) thresholds from 0.50 to 0.95 with a step size of 0.05, as well as top-$k$ classification accuracy (i.e., if the ground truth label appears in the $k$ labels with the highest probabilities). The results show that OCR leads to a significant improvement across all metrics, with mAP improving by a factor of 12 and top-1 accuracy improving by a factor of 3.

Table 5: Object Detection Refinement Results. We study the effect of OCR and report the mean average precision (mAP) of the predicted bounding boxes and top-K accuracy of the predicted labels.

| Method | mAP ($\uparrow$) | Top-K Accuracy % ($\uparrow$) | | |
|---|---|---|---|---|
| | | k=1 | k=3 | k=5 |
| ViLD | 2.4 | 14.7 | 32.3 | 41.6 |
| ViLD + OCR | 28.9 | 45.0 | 62.0 | 69.5 |

# B  Creating the Semantic Distribution

## B.1  Affinity Matrix Generation

We compare two ways to generate affinity matrices. First, we generate the affinity matrices using large language models using the following procedure: 1) We replace both the {obj} and the {target object} in the prompt: "I see the following in a room: {obj}. This is likely to be the closest object to {target object}." 2) We find the log probability of the target object instead of the {obj} since it will be represented by the same number of tokens regardless of the {obj} in this prompt and use this instead for representing affinity values. Second, we use an embedding model to encode {obs} and {target object} and use the cosine similarity to find the affinity values. We normalize when appropriate. For the pharmacy domain, in Table 6, we generate affinity matrices with different LLMs and embedding models and then compare the quality of affinity matrices quantitatively by comparing them to the open-source Google Product Taxonomy [110] as the "ground truth" matrix. Visualizations of the affinity matrices generated by the ground truth, the best embedding model (OpenAI Embeddings), and the best LLM are shown in Figure 3 for the pharmacy domain. In the pharmacy domain, we have the following 6 categories and items from the taxonomy:

1. Supplements: vitamins, fish oil, omega-3, calcium, probiotics, protein powder, COQ10, anthocyanin

2. Hair Care: shampoo, conditioner

3. Oral Care: toothpaste, toothbrush, dental floss

4. Cosmetics: face wash, sunscreen, lotion, hand cream, body wash

5. Medication: aspirin, tylenol, ibuprofen, advil, pain relief

6. Outliers: shaving cream, eye drops, deodorant, band-aid

For the ground-truth matrix, all elements in a category are given uniform affinities to each other, and each row is normalized to sum to 1.0 probability. Note that each item in the "outliers" category (e.g., eye drops) does not belong to any of the other 5 categories and is treated as its own category. We use the Google taxonomy to categorize the objects within each category. With the categories listed in order along both axes of the matrix, the ground truth affinity matrix has a block-diagonal structure with a uniform block for each category (Figure 3A). We evaluate the following LLMs and embedding models off-the-shelf, without finetuning: BERT [115], CLIP [27], embeddings from the OpenAI API [108], OPT-13B [116], and PaLM. JSD measures the similarity between two probability distributions, so we measure the similarity between each row of the affinity matrix and the corresponding row of the ground truth. Then, we average across the rows to get the average distance from each object's probability distribution to that object's ground truth. We observe that the choice of LLM has a significant impact on the affinity matrix (Table 6), and that the LLMs can approximately recover the block diagonal structure of the ground truth matrix (Figure 3). PaLM attains the highest accuracy, with a 44.6% improvement over a uniform affinity matrix.

Table 6: Affinity matrix results. We report the average Jensen-Shannon Distance (JSD) between each row of the affinity matrix and the ground truth matrix, as well as the percentage improvement over the uniform JSD (i.e., (uniform JSD - method JSD) / uniform JSD).

| Method | JSD ($\downarrow$) | % Improvement ($\uparrow$) |
|---|---|---|
| Uniform | 0.65 | N/A |
| BERT Embedding | 0.64 | 1.5 |
| CLIP Embedding | 0.52 | 20.0 |
| OpenAI Embedding | 0.43 | 33.8 |
| OPT-13B | 0.38 | 41.5 |
| **PaLM** | **0.36** | **44.6** |

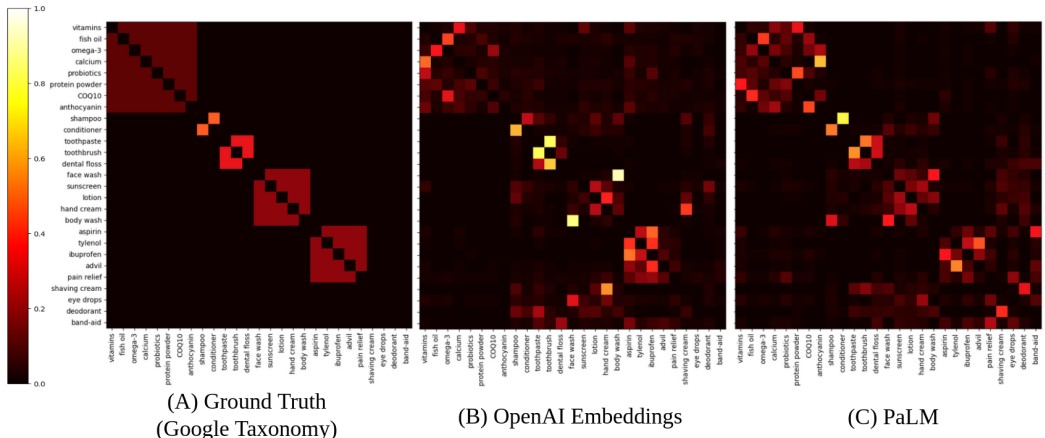

(A) Ground Truth
(Google Taxonomy)

(B) OpenAI Embeddings

(C) PaLM

Figure 3: Three different affinity matrices for the pharmacy domain. The left shows the affinity matrix generated from the Google Taxonomy. The center shows the affinity matrix generated by the OpenAI Embeddings model and the right shows the matrix generated by PaLM.

## B.2 Offline Semantic Distribution Generation with Object List

We now generate a semantic distribution based on the affinity matrix and detected objects. The semantic occupancy distribution models the probability that the target object occupies a given location, given the classes of observed objects in the scene, i.e. $P(L_T = l \mid L_{1...n} = l_{1...n}, C_{1...n} = c_{1...n})$, where $L_T$ is the location of the target object, $L_{1...n}$ are the positions of the *visible* objects, and $C_{1...n}$ are the inferred classes of the visible objects. We abbreviate this quantity as $P(L_T = l \mid L, C)$.

We interpret affinity values $M_{ij}$ to be the probability of object $j$ being the closest to object $i$ in expectation across scenes. However, given the current scene, there may be more or less space that is nearest to a particular object, so we interpret these affinity values as being normalized per unit area. Thus, formally, given $N(l)$ representing the index of the object closest to location $l = (x_l, y_l)$, $P(L_T = l \mid L, C) \propto M_{target, N(l)}$.

In simulation experiments for constrained environments, $N(\cdot)$ is computed using the 3D coordinates of the visible objects obtained from the depth image. We compute the 2D semantic occupancy distribution (in the horizontal plane of the shelf) and reduce it to 1D by summing along camera rays. In physical experiments, to avoid errors due to noisy depth readings we compute the distribution directly in 2D, using pixel distance for $N(\cdot)$ instead of world coordinates.

## C  Closed-World Downstream Mechanical Search Policies

### C.1  Problem Statement

We consider the problem of robotic mechanical search for a target object $\mathcal{O}_T$ in a cluttered, semantically organized shelf containing the target and $N$ other rigid objects $\{\mathcal{O}_1, ..., \mathcal{O}_N\}$ of cuboidal shapes in stable poses. We build on the problem statement and assumptions in Huang et al. [4]. We model the setup as a finite-horizon Partially Observable Markov Decision Process (POMDP). States $s_t \in \mathcal{S}$ consist of the full geometries and poses of the objects in the shelf at timestep $t$ and observations $y_t \in \mathcal{Y} = \mathcal{R}^{H \times W \times 4}$ are RGBD images from a robot-mounted depth camera at timestep $t$. Actions $a_t \in \mathcal{A} = \mathcal{A}_p \cup \mathcal{A}_s$ are either *pushing* or *suction* actions, where the former are horizontal linear translations of an object along the shelf and the latter pick up an object with a suction gripper and translate it to an empty location on the shelf with no other objects in front of it. We make the following assumptions:

- The dimensions of the shelf are known.
- Each dimension of each object is between size $S_{min} = 5\,cm$ and size $S_{max} = 25\,cm$.

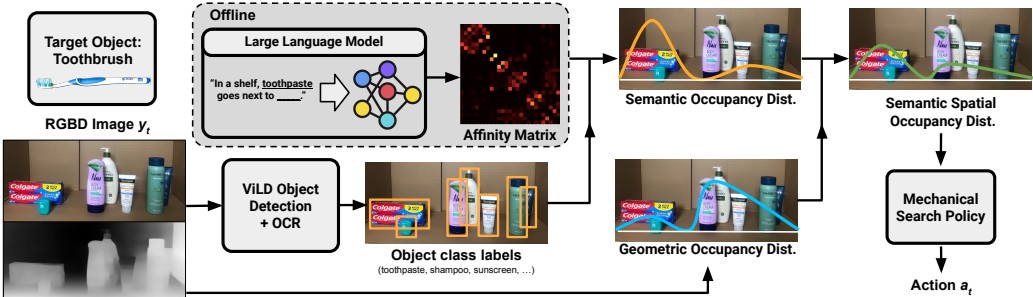

Figure 4: **Overview of SMS for Constrained Environments with Downstream Mechanical Search Policy.** SMS first receives a scene image and a desired target object. Since the object list is known, it then applies object detection and OCR to identify objects within the scene. SMS then uses a large language model to compute affinities between detected objects to the target object, and it uses these affinities to output a semantic occupancy distribution of the appropriate dimension for the downstream problem. This distribution indicates the likelihood of the physical presence of objects which is used to determine the next action by the downstream mechanical search policy.

- The shelf is semantically organized.
- The names of all objects in the shelf are a subset of a known list of object names.
- Actions cannot inadvertently topple objects or move multiple objects simultaneously.

## C.2 LAX-RAY

LAX-RAY[4] is a mechanical search policy for shelf environments. LAX-RAY have utilized geometric information by considering object geometries and camera perspective (e.g., tall target objects cannot be occluded by short objects and objects in the center of an image occlude more areas) to facilitate the search. It consists of a perception module and a greedy action selection module. The perception module takes the depth observation and predicts the geometric/spatial occupancy distribution to encode the geometric information. LAX-RAY learns this module on a simulation dataset, with the ground-truth occupancy distribution calculated using Minkowski sum. A greedy action selection module called Distribution Area Reduction (DAR) selects robot actions to greedily reduce the overlap between objects and the distribution. Another search policy has been proposed in [3], namaed Distribution Entropy Reduction (DER). DER selects the action that would reduce the entropy of the distribution the most after taking the action. We denote the searching pipeline with DER to be LAX-RAY (DER).

We show the SMS pipeline specifically for constrained environments with LAX-RAY in Figure 4. This pipeline was used to conduct the simulation mechanical search experiments in Section D.2.1 and the physical mechanical search experiments in Section D.4.

## D Experiments

### D.1 Scene Generation

The taxonomy defines a tree where each category is a node and each object name is a leaf node. To create a scene with $N$ objects in a given domain, we begin by uniformly sampling $N$ objects without replacement from the total objects available in that domain. We then generate scenes in a top-down recursive manner using the taxonomy tree. At the root, we start with the whole shelf available to us. At each node, we split the shelf in half either horizontally or vertically with 50% probability each and recursively continue scene generation in these sub-shelves. If a node has more than 8 descendants, however, we always split the scene horizontally to avoid overcrowding resulting from the aspect ratio of the shelf. At each level of recursion, we accumulate random noise to the eventual placement of each object in the current branch, uniformly sampled from -2 cm to 2 cm. At the last non-leaf node, we place all leaves in random positions within the current level's sub-shelf.

Table 7: Simulation Experiment Results.

**Pharmacy Domain**

| | 12 objects | | 15 objects | | 18 objects | | 21 objects | |
|---|---|---|---|---|---|---|---|---|
| | **Successes** | **# Actions** | **Successes** | **# Actions** | **Successes** | **# Actions** | **Successes** | **# Actions** |
| **LAX-RAY** | 168/190 | $4.06 \pm 0.23$ | 160/186 | $5.17 \pm 0.28$ | 144/188 | $5.78 \pm 0.44$ | 104/177 | $8.24 \pm 0.67$ |
| **SMS-E** | **176/190** | $2.90 \pm 0.18$ | 159/186 | $3.77 \pm 0.26$ | 146/188 | $5.05 \pm 0.42$ | 110/177 | $5.69 \pm 0.54$ |
| **SMS-LLM** | **176/190** | $\mathbf{2.66 \pm 0.14}$ | **162/186** | $\mathbf{3.26 \pm 0.19}$ | **150/188** | $\mathbf{4.25 \pm 0.34}$ | **118/177** | $\mathbf{5.47 \pm 0.43}$ |

**Kitchen Domain**

| | 12 objects | | 15 objects | | 18 objects | | 21 objects | |
|---|---|---|---|---|---|---|---|---|
| | **Successes** | **# Actions** | **Successes** | **# Actions** | **Successes** | **# Actions** | **Successes** | **# Actions** |
| **LAX-RAY** | 185/192 | $2.15 \pm 0.14$ | 182/194 | $2.97 \pm 0.23$ | 177/193 | $3.99 \pm 0.29$ | 159/191 | $4.36 \pm 0.38$ |
| **SMS-E** | **186/192** | $\mathbf{1.56 \pm 0.08}$ | **188/194** | $2.15 \pm 0.15$ | **184/193** | $3.00 \pm 0.27$ | **167/191** | $3.07 \pm 0.25$ |
| **SMS-LLM** | 184/192 | $1.60 \pm 0.10$ | 184/194 | $\mathbf{2.04 \pm 0.13}$ | 179/193 | $\mathbf{2.97 \pm 0.26}$ | 163/191 | $3.17 \pm 0.28$ |

**Office Domain**

| | 12 objects | | 15 objects | | 18 objects | | 21 objects | |
|---|---|---|---|---|---|---|---|---|
| | **Successes** | **# Actions** | **Successes** | **# Actions** | **Successes** | **# Actions** | **Successes** | **# Actions** |
| **LAX-RAY** | 172/194 | $2.60 \pm 0.18$ | 152/188 | $4.15 \pm 0.38$ | 136/190 | $4.64 \pm 0.37$ | 115/181 | $5.86 \pm 0.56$ |
| **SMS-E** | **173/194** | $3.01 \pm 0.22$ | 152/188 | $3.80 \pm 0.31$ | 140/190 | $4.78 \pm 0.44$ | 115/181 | $\mathbf{5.33 \pm 0.50}$ |
| **SMS-LLM** | 172/194 | $\mathbf{2.33 \pm 0.13}$ | **161/188** | $3.50 \pm 0.31$ | **142/190** | $\mathbf{3.75 \pm 0.32}$ | **123/181** | $5.50 \pm 0.49$ |

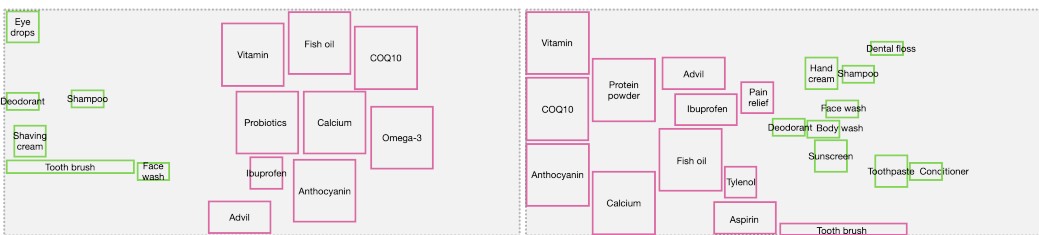

Figure 5: Two examples of layouts for the simulated scenes from the birds-eye view, where the top is the rear of the shelf and the bottom is the front. The layouts are generated from the procedure in Section D.1. The left scene is an example of a layout for a scene with 15 objects and right scene is a layout for a scene with 21 objects. Green rectangles represents objects in personal care categories and pink rectangles represents objects in supplements.

We resolve collisions by iteratively moving objects along the displacement vector between colliding objects and discard scenes where such a procedure takes longer than 1 second to run. We also discard scenes where there is no potential target object that is invisible from the camera's perspective at the start of the rollout. We reiterate that the taxonomy is *independent* of the language models used to generate affinities. The LLMs are applicable beyond manual semantic categorizations like the Google Taxonomy, but we use this resource for evaluation purposes. The scenes for all simulation, physical, and object detection experiments are generated by this procedure.

We use approximate sizes of these items to generate collision-free scenes. In simulation, we also scale these objects down in order to be able to run experiments on the same-sized shelf, which has an effect similar to running experiments in a larger shelf where more items could originally fit. The scaling factors for the pharmacy and kitchen domains are 0.7, but 0.4 in the office domain due to overall larger objects unable to easily fit and move within a small shelf.

## D.2 Simulation Experiments

### D.2.1 Simulation Experiments with LAX-RAY

We run an extensive suite of experiments using the same simulator as prior work in mechanical search on shelves [3] and study the benefit brought by SMS. We use a grid search on the average number of actions required in the pharmacy domain with 15 objects to tune the Gaussian smoothing $\sigma$ to be 50 pixels. We use the same parameters for the other two domains.

We generate scenes with various numbers of objects: $N = 12, 15, 18,$ and 21. We generate 200 scenes for each value of $N$. In Figure 5, we show example layouts of the scenes as created by the procedure in Section D.1. We discard scenes where the target object starts out visible, resulting in

just under 200 scenes for each value of $N$. Termination occurs when at least $X = 1\%$ of the target object becomes visible or reaching maximum action number $2N$. The reason for the low threshold is that the DAR policy has trouble making progress on a partially revealed target object [3], which may dilute the comparison between different methods for generating semantic distributions.

We report results for all numbers of objects $N$ in Table 7, SMS-LLM outperforms both SMS-E, while also beating LAX-RAY across various values of $N$ in terms of success rate (by an additional 30/741 scenes) and average number of actions required (by 32.4%). A point of note is that the action differential percentage grows as the number of objects increases. At 21 objects, LAX-RAY requires 8.24 actions on average, whereas SMS requires just 5.47. This trend agrees with intuition that it is unscalable to search large environments with no semantic intuition.

## D.3 Simulation Experiments with Object Detection Noise

| Method | No noise | 10% Noise | 50% Noise | 90% Noise | LAX-RAY |
|---|---|---|---|---|---|
| # of Actions | $3.81 \pm 0.31$ | $4.20 \pm 0.38$ | $4.44 \pm 0.41$ | $4.83 \pm 0.47$ | $5.12 \pm 0.43$ |

Table 8: Experiment to determine the impact of object detection noise on task performance (# of actions). For SMS-LLM, we randomly perturb the object detection (i.e. randomly select a label from the object list) with probability P. We do 400 rollouts over the categories of 12, 15, 18, 21 objects in the scene for the pharmacy domain. We report the average number of actions taken to reveal the target object and standard error. We see the general trend as object detection noise increases the task performance decreases.

We study the impact of the object detection accuracy on the task performance. We randomly change the object labels with a probability $P$. The results are shown in Table 8, where $P = 0.1, 0.5, 0.9$. The number of actions needed to find the occluded object increases as $P$ increases. This is because random perturbations can cause the semantic distribution to approach a uniform distribution thus not modifying the existing action of the downstream policy. Therefore, Table 8 indicates there is also a strong positive correlation between object detection accuracy and task performance.

### D.3.1 Simulation Experiments with DER

| Policy | 12 objects | | 15 objects | | 18 objects | | 21 objects | |
|---|---|---|---|---|---|---|---|---|
| | Success | # Actions | Success | # Actions | Success | # Actions | Success | # Actions |
| LAX-RAY (DER) | 84% | 5.79±0.38 | 74% | 7.69±0.54 | 62% | 8.08±0.64 | 42% | 9.52±0.72 |
| SMS-LLM | 90% | 4.42 ± 0.39 | 81% | 5.06±0.43 | 71% | 7.11±0.60 | 45% | 6.87±0.67 |

Table 9: Simulation experiments results of SMS-LLM with DER for the Pharmacy domain. We ablate the downstream policy and see that SMS-LLM outperforms LAX-RAY with DER. We report the number of rollouts that were successful and the mean actions to retrieve the occluded object and the standard error.

We integrate SMS with a different downstream policy Distribution Entropy Reduction (DER) from [3]. We multiply the semantic distribution with the geometric distribution as the input to DER. DER selects the action minimizing the distribution entropy after taking the action. We use the same setup and scenes as in Section D.2.1. We report the results for 100 scenes with 12 objects in Pharmacy domain in Table 9.

## D.4 Physical Experiments

We generate the physical scenes with the scene generation procedure outlined in Section D.1 to ensure the scenes were not biased. Examples of physical environment layouts are shown in Figure 6.

Because the RealSense camera is not able to capture the fine details of the text on the objects when observing the entire scene at resolution $640 \times 480$ pixels, we perform a three-stage scan of the scene by moving the end-effector to 3 adjacent positions, all of which are closer to the shelf, where the text is more easily readable. At each of these poses, we take a picture of the scene, project the known world position of the objects to the new camera frame, identify text with OCR, and assign each text detection to the object it is contained in. If there are detections on the same object from multiple scan locations, we use the OCR that has the lowest entropy for its distribution, a measure of confidence. During the physical experiments rollouts, when the action given by the policy causes

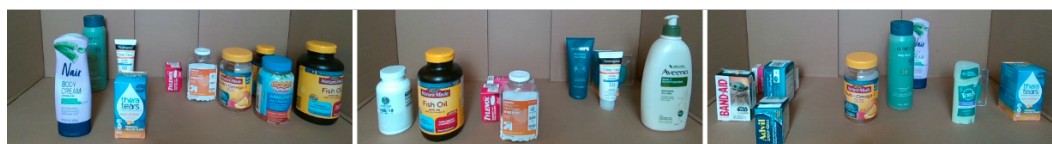

Figure 6: Here are 3 example scenes from the physical mechanical search experiments in the constrained environment setting.

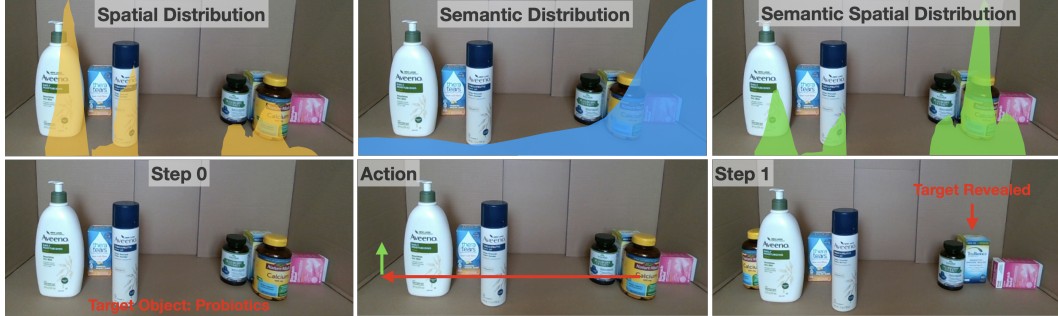

Figure 7: Physical rollout example with the target object being the probiotics. **Top:** the spatial distribution, semantic distribution and semantic spatial distribution for step 0. **Bottom:** RGB observations at step0, the action given by DAR and the RGB observation after executing the action.

unintentional toppling or a missed grasp due to depth sensor noise, we reset the object to undo the action and run the policy again.

We show a physical experiment rollout with the target object being the probiotics as in Figure 7. In this rollout, the spatial distribution generated based on geometric information by LAX-RAY indicates the left side of the shelf occludes more area. However, the semantic distribution generated by SMS indicates the target object is more likely to be on the right. This is because other objects from the supplements category where the target object probiotics belongs to are visible on the right. Combining the spatial distribution and semantic distribution into the semantic spatial distribution takes into account both the geometry and semantic information and results in a more accurate distribution.

### D.4.1 Ablating Object Lists for Semantic Distribution for Physical Experiments

| Metric | Uniform Dist. | SMS-LLM w/o Object List | SMS-LLM w/ Object List |
|---|---|---|---|
| JSD ↓ | $0.554 \pm 0.006$ | $0.421 \pm 0.032$ | $0.382 \pm 0.036$ |

Table 10: We measure the deviation in the semantic distribution generated by these methods and the ground truth using JSD. SMS with the object list, which uses Object Detection+OCR, outperforms SMS without the object list, which uses the Crop Generation + Image Captioning pipeline.

To evaluate the benefit of object lists, we compare the performance of our method on shelves with and without access to the object list by computing the Jensen-Shannon Distance (JSD) [109] between the generated distribution and the ground truth distribution on the 12 physical shelves as in Section 5.2.2. From Table 10, we see that SMS achieves a better semantic distribution compared to a uniform prior in both cases. The SMS-LLM w/o Object List uses the same Crop Generation + Image Captioning pipeline as the open-world experiments which is equivalent to using a VLM for scene understanding. The SMS-LLM w/Object List uses the object list for object detection and to refine the labels using OCR. We see that knowing the object list improves results, which is expected as it reduces the noise in the scene understanding and leads to a higher quality of the semantic distribution.

### D.5 Experiments for Open-World Environments

We also ablate the modules in SMS-E with results shown in Table 11, where we study the impact of using CLIP weighting, different image captioning models and SAM crops. As mentioned previously that image captioning can be noisy, we use CLIP to verify the captions. We refer this as CLIP weighting. Without this, the performance drops by 21%. When we use BLIP-IC instead of BLIP-2

| Ablations | w/o CLIP Weighting | BLIP-IC | w/o SAM | SMS-E |
|---|---|---|---|---|
| IoU | 0.307± 0.038 | 0.310 ± 0.043 | 0.286± 0.038 | **0.391**± 0.039 |

Table 11: IoU results for ablated SMS-E. **w/o CLIP Weighting** doesn't using CLIP to refine the generated captions as described in Section 4.2. **BLIP-IC** use BLIP-IC to get the descriptions for each crops instead of BLIP-2. BLIP-IC is linked in the Appendix Section D.5. **w/o SAM** doesn't use crops given by SAM and crops generated by multi-scale sliding windows are used.

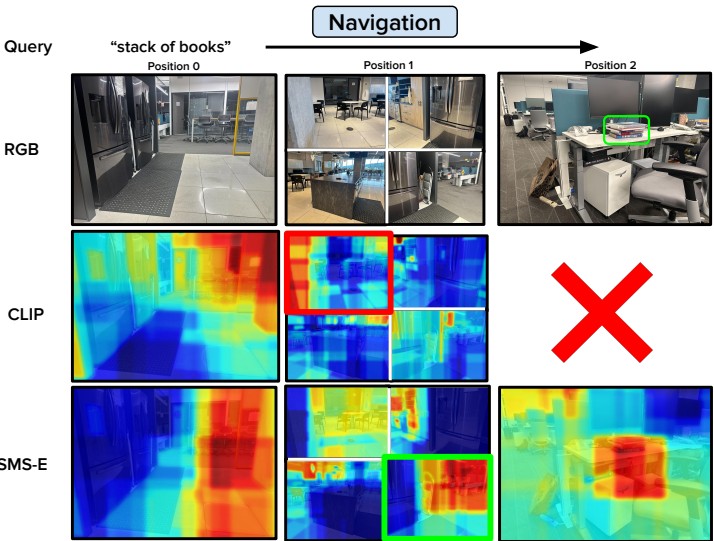

Figure 8: **Object navigation experiment in BAIR Office Kitchen.** A short horizon navigation example where we start at position 0 and end at position 2. SMS is able to correctly maneuver to the stack of books while CLIP fails because its bag-of-words nature is susceptible to incorrectly assigning high probability to the pillars in the scene as they are semantically related to "stack."

for image captioning, the performance drops by 21%. Finally, without cropping using SAM to get object-centered crops, the performance drops by 27%.

More examples of the semantic distribution comparison between SMS and CLIP-based models are shown below. BLIP-IC is the large image captioning model of BLIP.

We also conduct a preliminary navigation experiment where a mobile robot follows the downstream navigation policy described in Section 4.3 and selects the physical location to move to based on the semantic distribution from CLIP and SMS-E. As shown in Figure 8, CLIP makes an incorrect turn (at position 1 it continues in the direction of the view with the red box) because of its bag-of-words behavior and attributes "stack of books" to having higher semantic similarity to concrete pillars in the scene rather than the area with office desks and chairs. SMS-E continues towards the office (green box in position 1) and finds a stack of books on a desk successfully.

## D.6 Object Lists in Closed-World Environments

**Pharmacy Domain** : vitamins , fish oil , omega-3 , calcium , probiotics , protein powder , COQ10 , anthocyanin , shampoo , conditioner , toothpaste , toothbrush , dental floss , face wash , sunscreen , lotion , hand cream , body wash , aspirin , tylenol , ibuprofen , advil , pain relief , shaving cream , eye drops , deodorant , band-aid

**Kitchen Domain** : spoon , ladle , spatula , tongs , whisk , fork , peeler , grater , saucepan , frying pan , salt , pepper , cumin , coriander , basil , turmeric , parsley , oregano , sugar , flour , cornstarch , oats , quinoa , rice

**Office Domain** : pen , pencil , highlighter , sticky note , binder paper , printer paper , index card , paper clip , rubber band , stapler , staples , tape dispenser , 3-hole punch , dry erase marker , sharpie ,

label maker , notebook , eraser , white-out , calculator , thumbtack , pencil sharpener , bubble wrap , styrofoam , packing tape , shipping boxes , ethernet cable , modem , router , network card , network bridge , headphones , speakers , aux cable , microphone , keyboard , mouse , USB adapter , hard drive , flash drive

# E    Object Lists and Examples for Open-World Environments

In this section, we show more examples of the semantic distributions generated by different methods from the static dataset and all the scenes.

**Grocery Object list**: 'blender', 'juicer', 'spatula', 'spray tan', 'sunglasses', 'gardening gloves', 'grass seeds', 'headphones', 'pruning shears', 'SD card', 'crayons', 'paper towel', 'plunger', 'Powerade', 'router', 'bottle opener', 'corkscrew', 'Danimals', 'Paneer', 'yogurt', 'bagel', 'baguette', 'daisy', 'danish pastry', 'red rose', 'toaster strudel', 'cocoa powder', 'incense sticks', 'succulents', 'condensed milk', 'kale', 'lotion', 'scotch tape', 'garlic bread', 'moisturizing masks'

**Office Object list** : 'box of paper', 'cat food', 'ice', 'leftover meatloaf', 'three ring binder', 'Budweiser beer', 'coke can', 'lion figurine', 'tequila', 'panda soft toy', 'acetone', 'facial cotton pad', 'HDMI cable', 'lipstick', 'pillow', 'beef patties', 'expo marker', 'mayo', 'relish', 'USB flash drive', 'thumb tacks', 'chain', 'Pringles', 'trail mix', 'iPhone charger', 'throw blanket', 'shears', 'emergency whistle', 'office party notice', 'yogurt', 'napkin holder', 'Academy Award'

**Home Object list**  : 'bar soap', 'boarding pass', 'empty bottle', 'pajamas', 'toothpaste', 'microphone', 'mail', 'laundry brush', 'paint roller', 'hand wraps', 'paddle', 'alarm clock', 'duvet', 'medal', 'pool balls', 'pool rack', 'soap', 'tissues', 'Neosporin', 'HP scanner', 'resistance bands', 'jump rope'

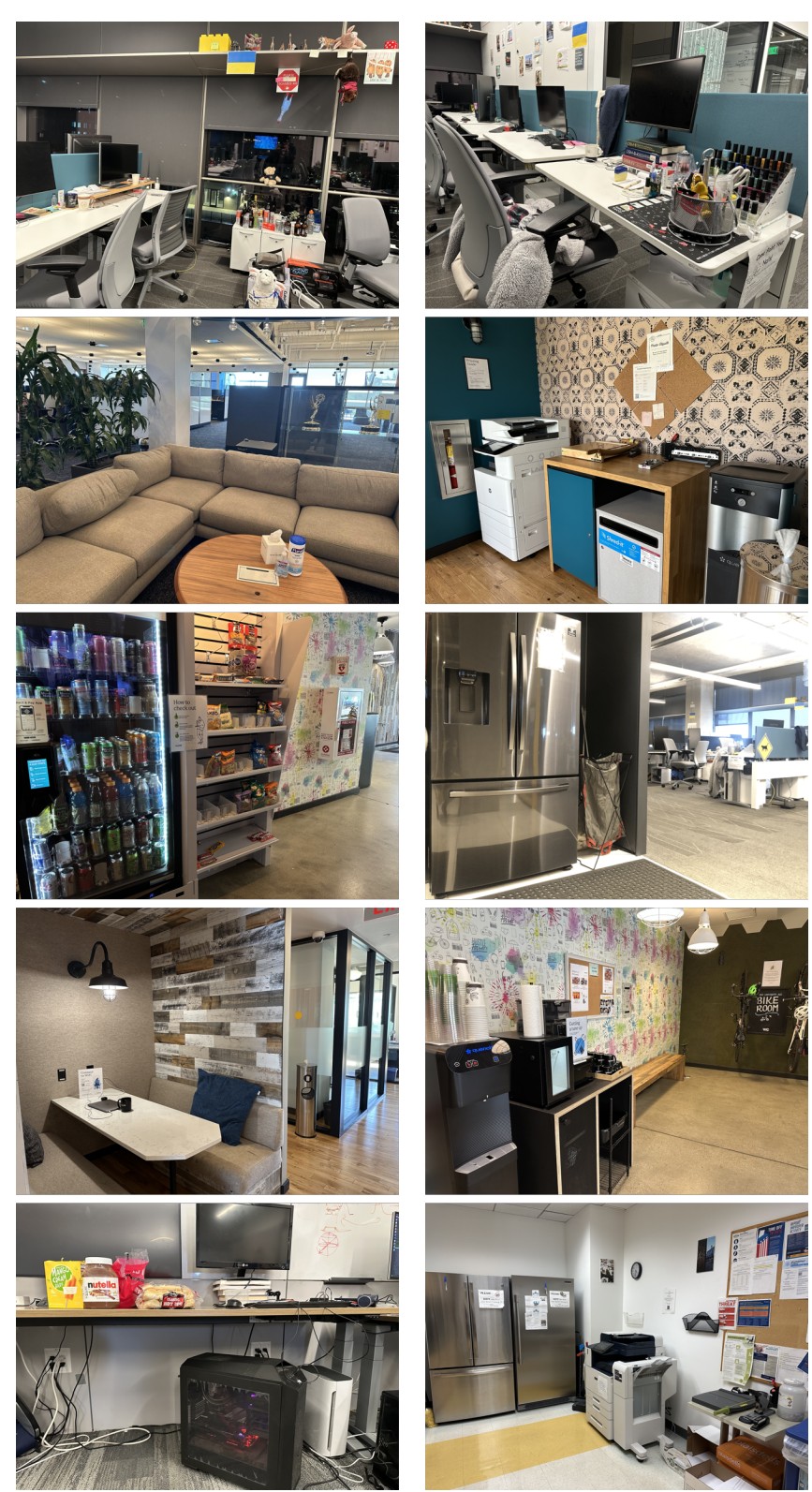

Figure 9: Office environments.

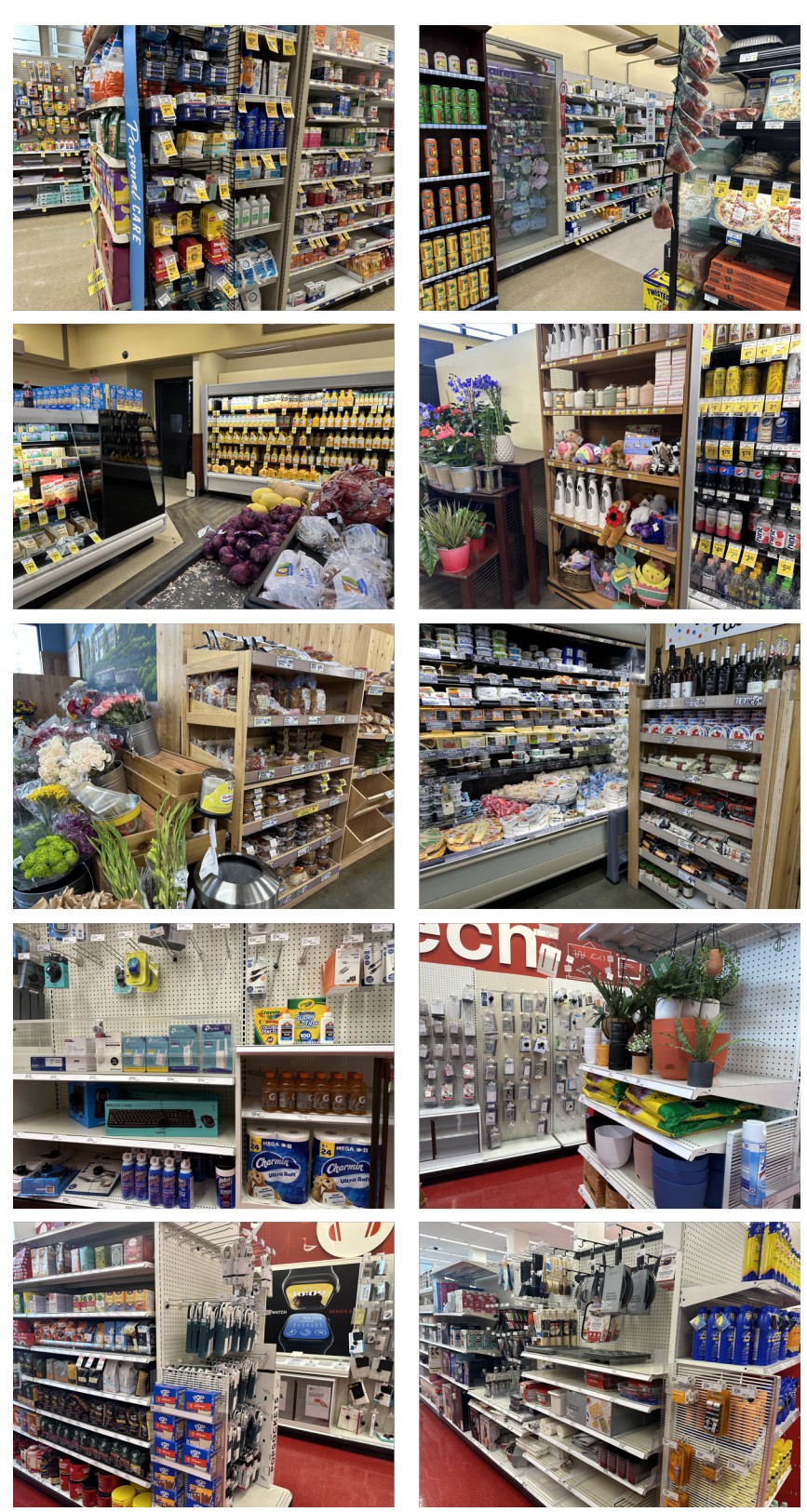

Figure 10: Grocery stores environments.

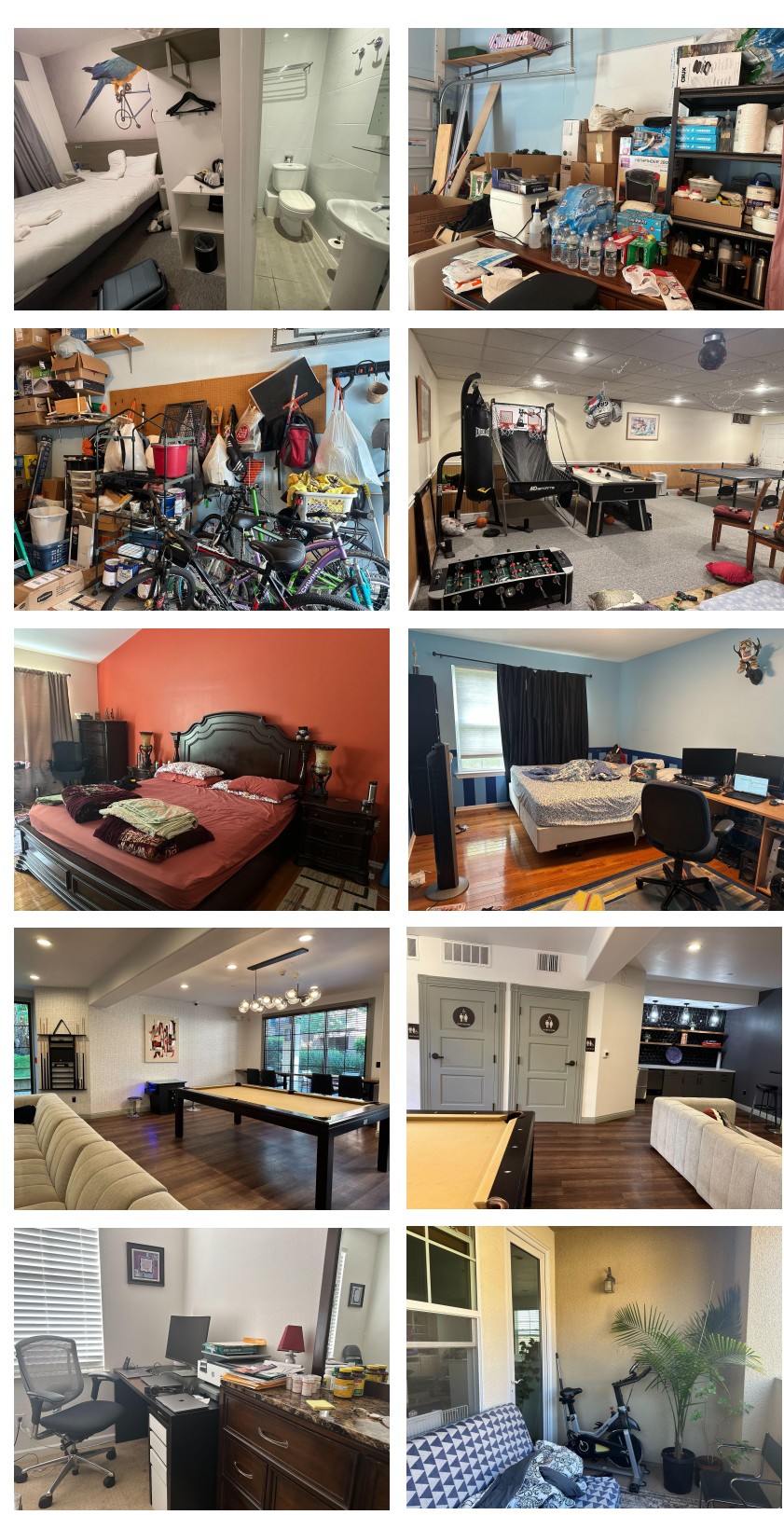

Figure 11: Home environments.

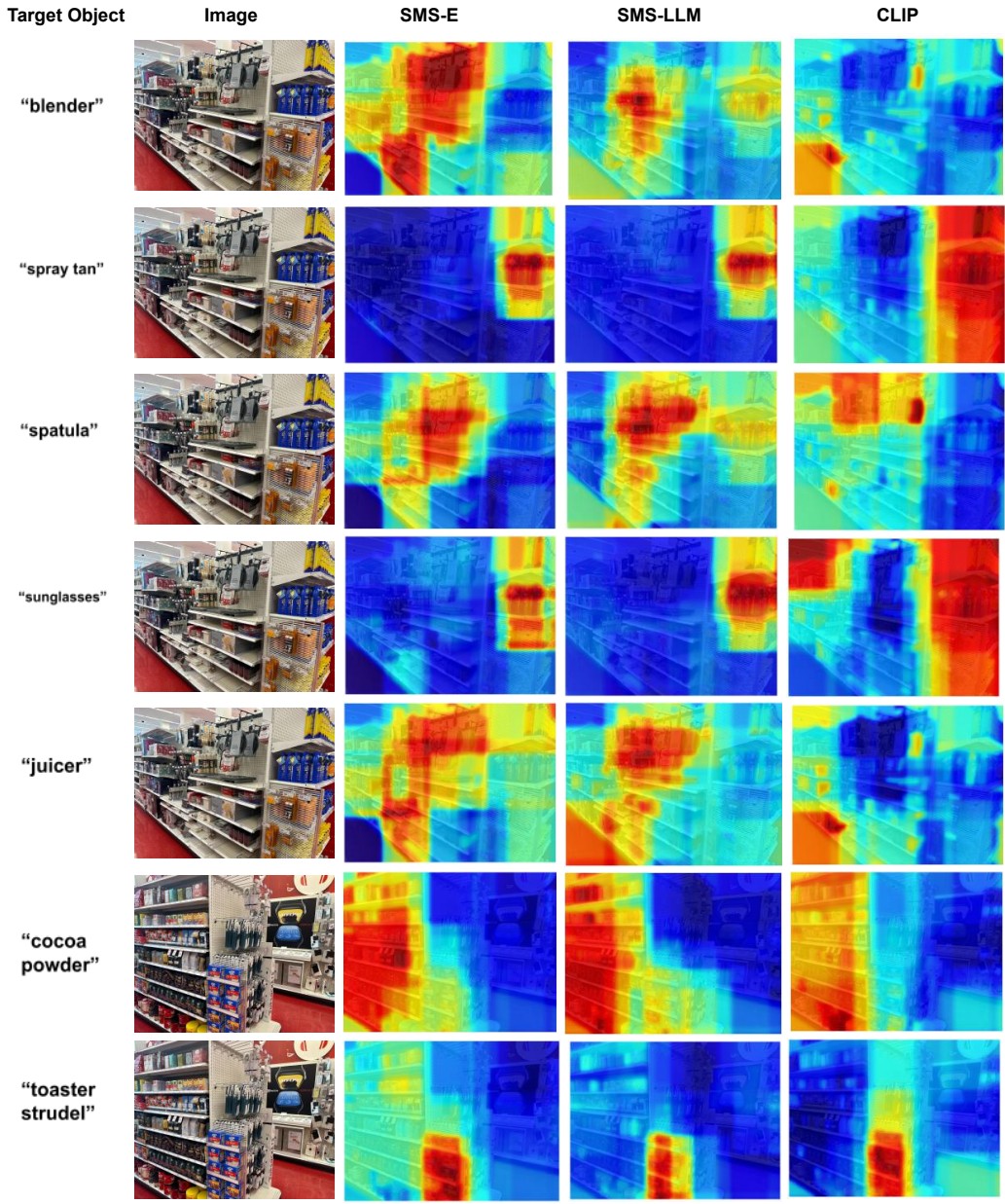

Figure 12: Example set 1 of the semantic distributions generated by different methods for grocery stores.

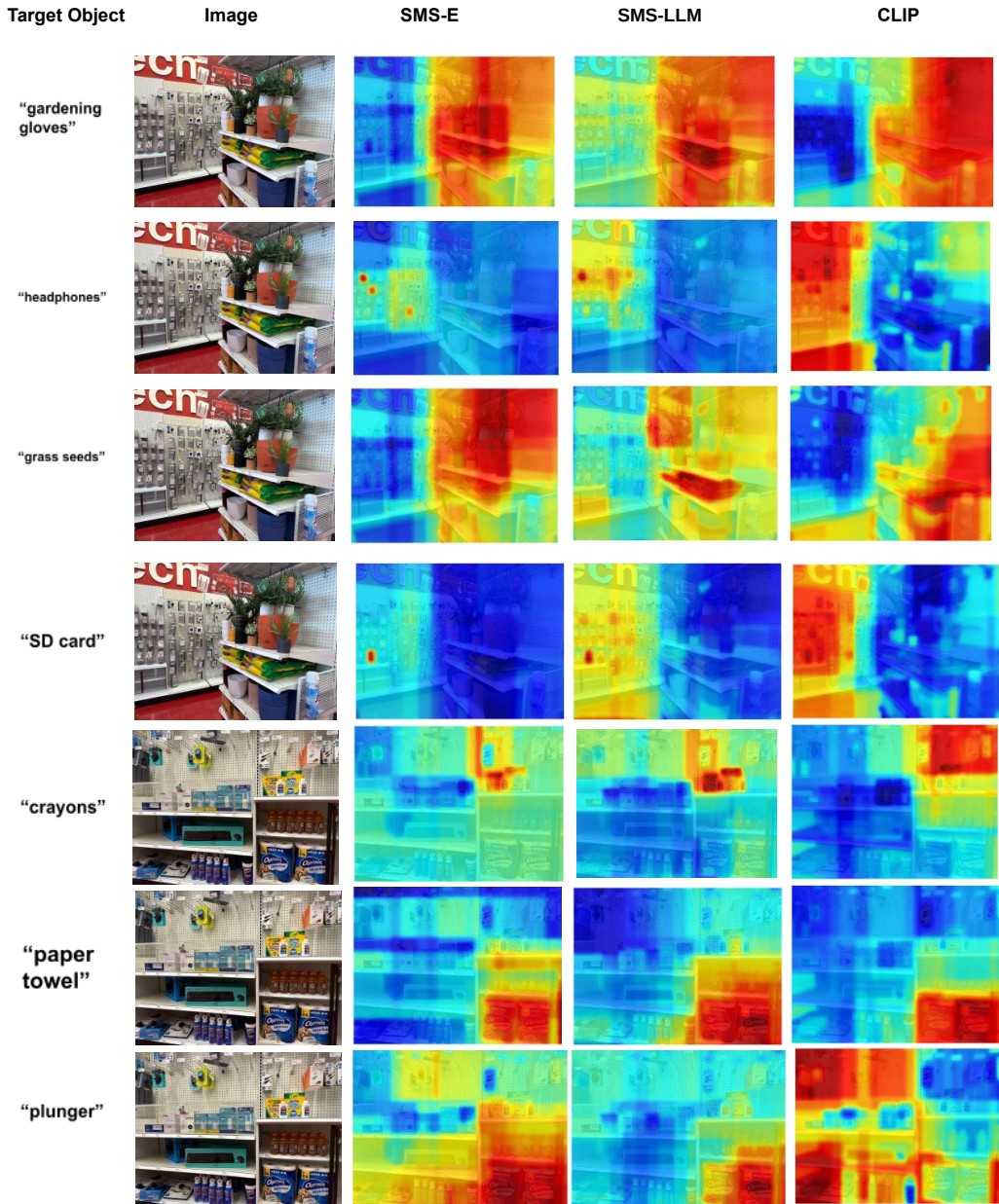

Figure 13: Example set 2 of the semantic distributions generated by different methods for grocery stores.

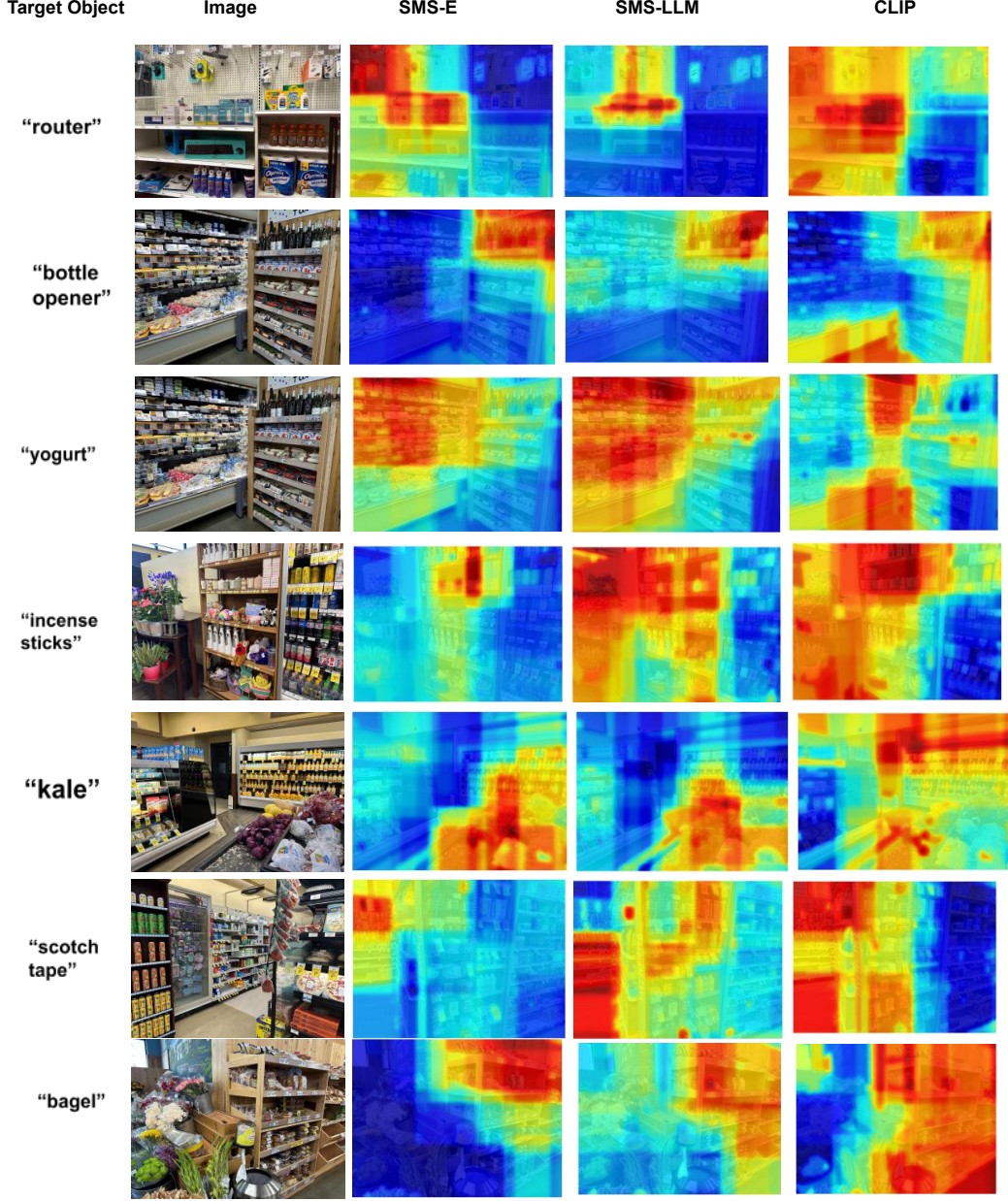

Figure 14: Example set 3 of the semantic distributions generated by different methods for grocery stores.

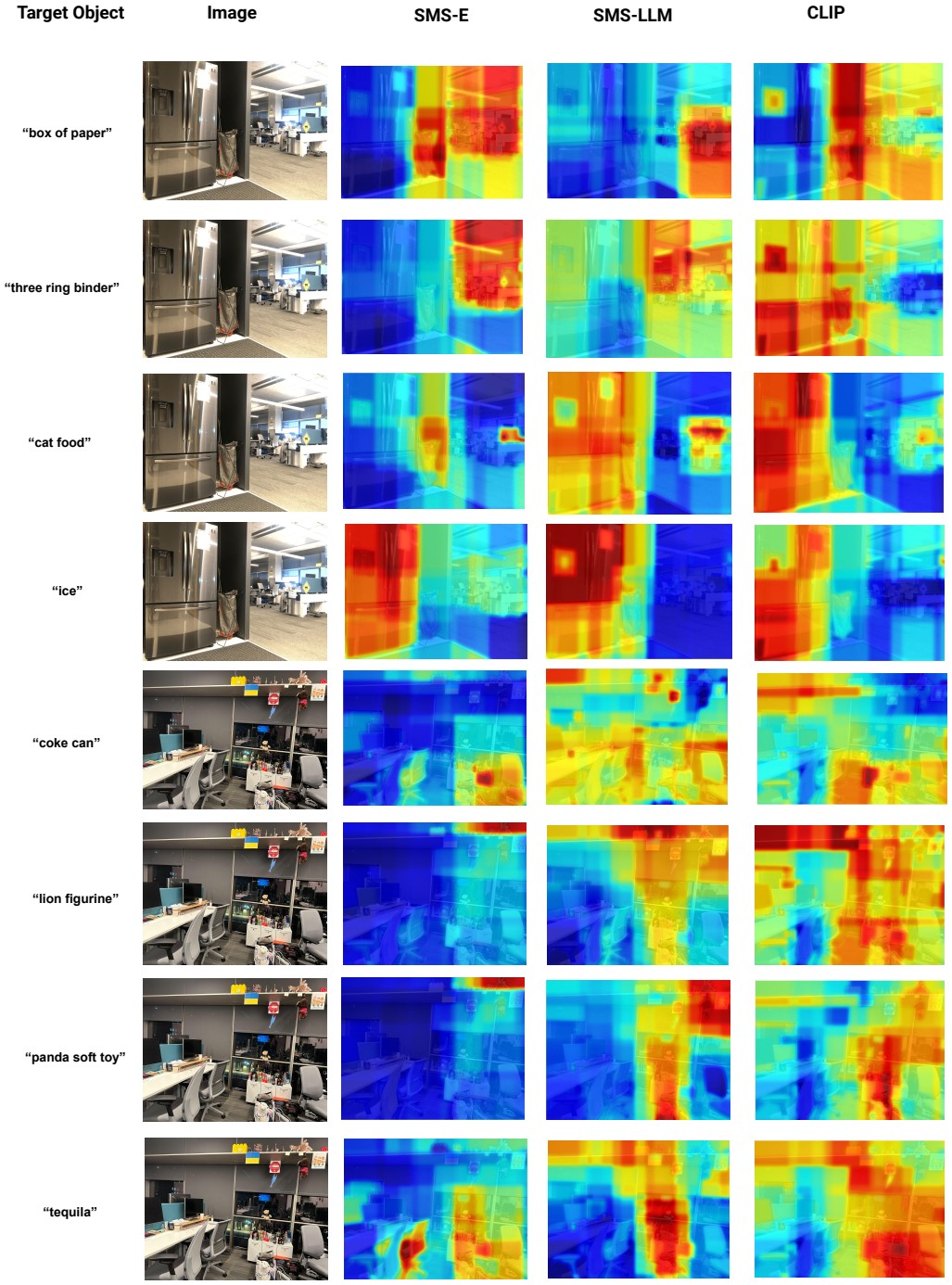

Figure 15: Example set 1 of the semantic distributions generated by different methods for offices.

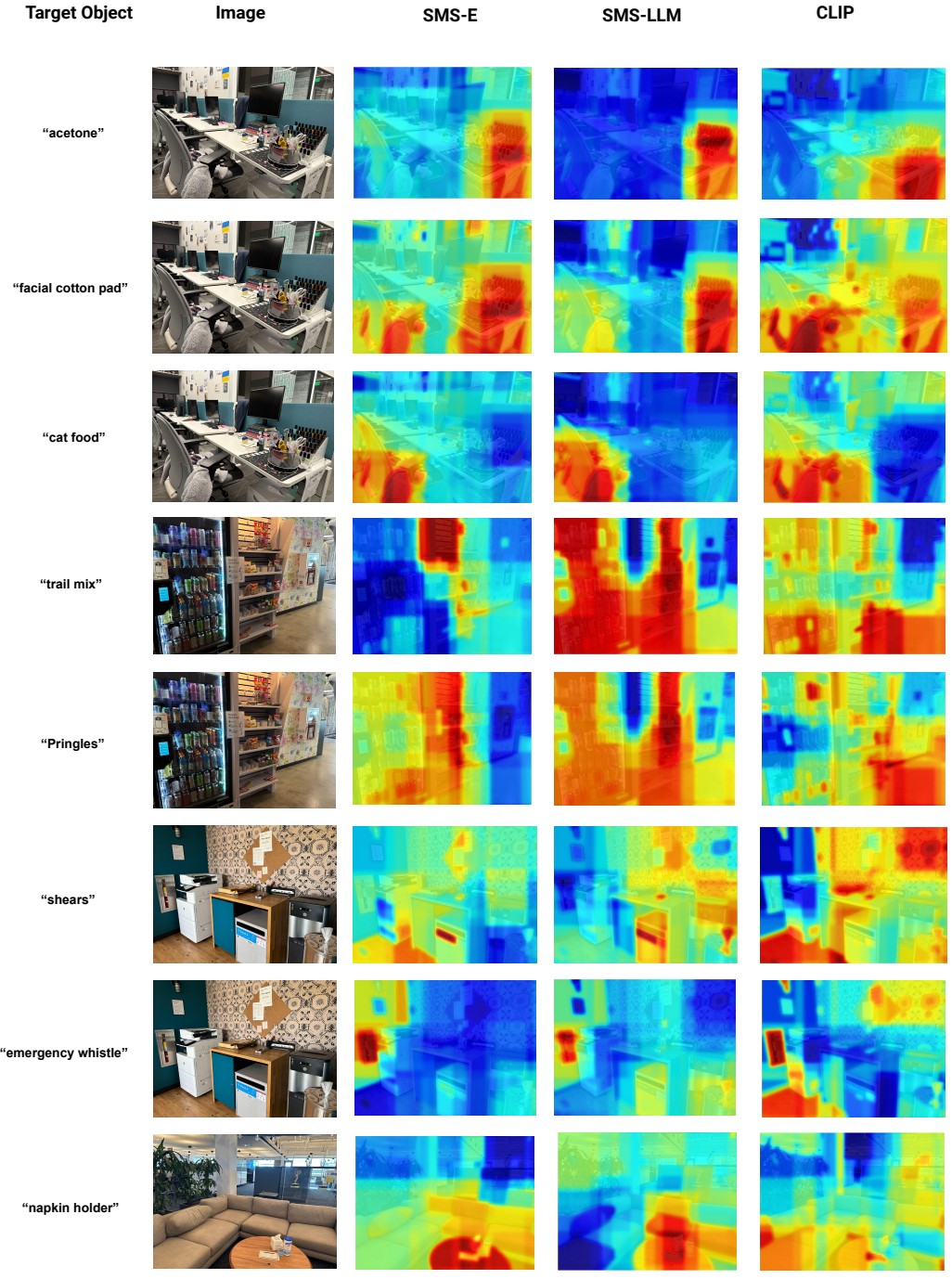

Figure 16: Example set 2 of the semantic distributions generated by different methods for offices.

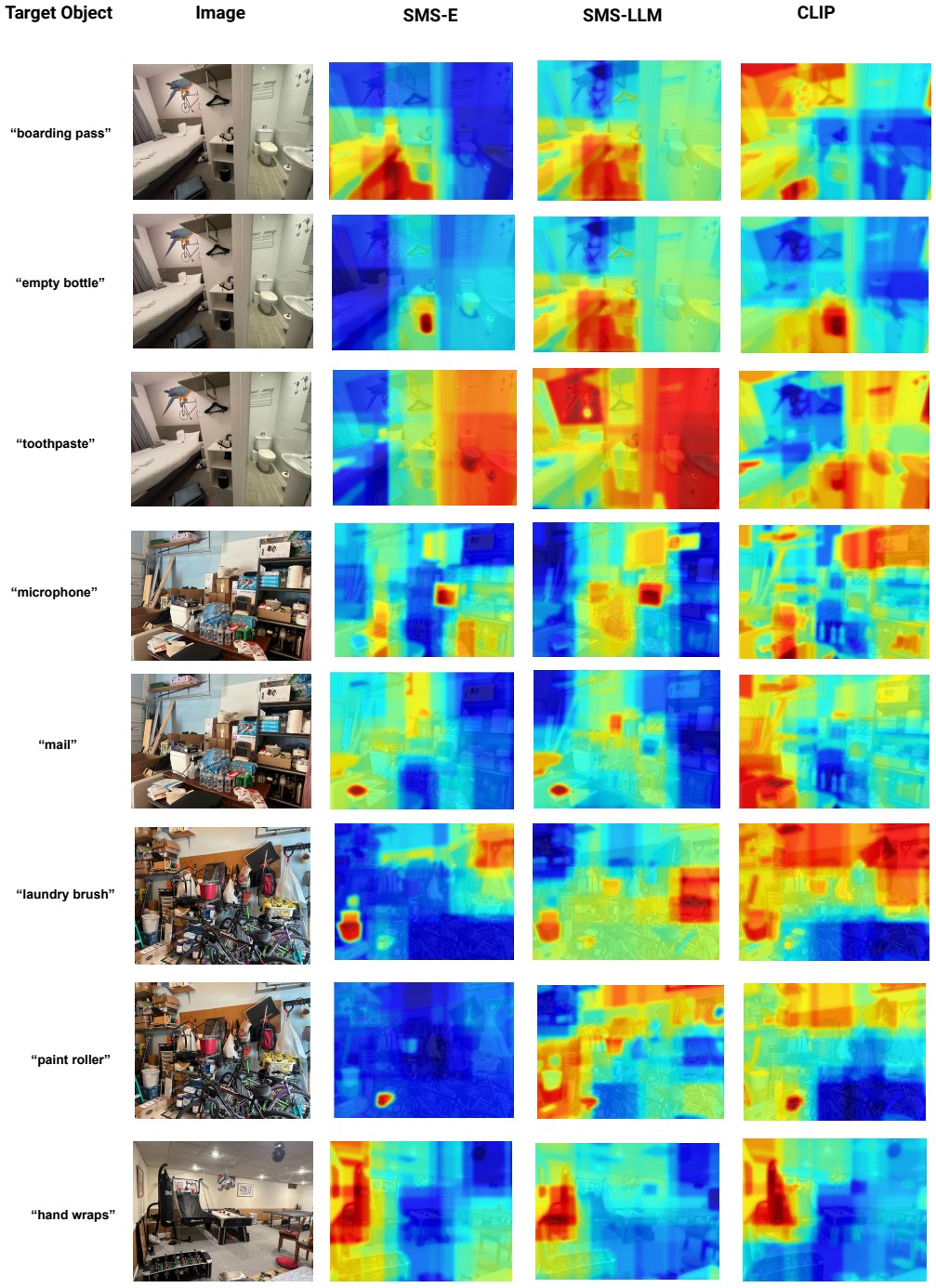

Figure 17: Example set 1 of the semantic distributions generated by different methods for houses.

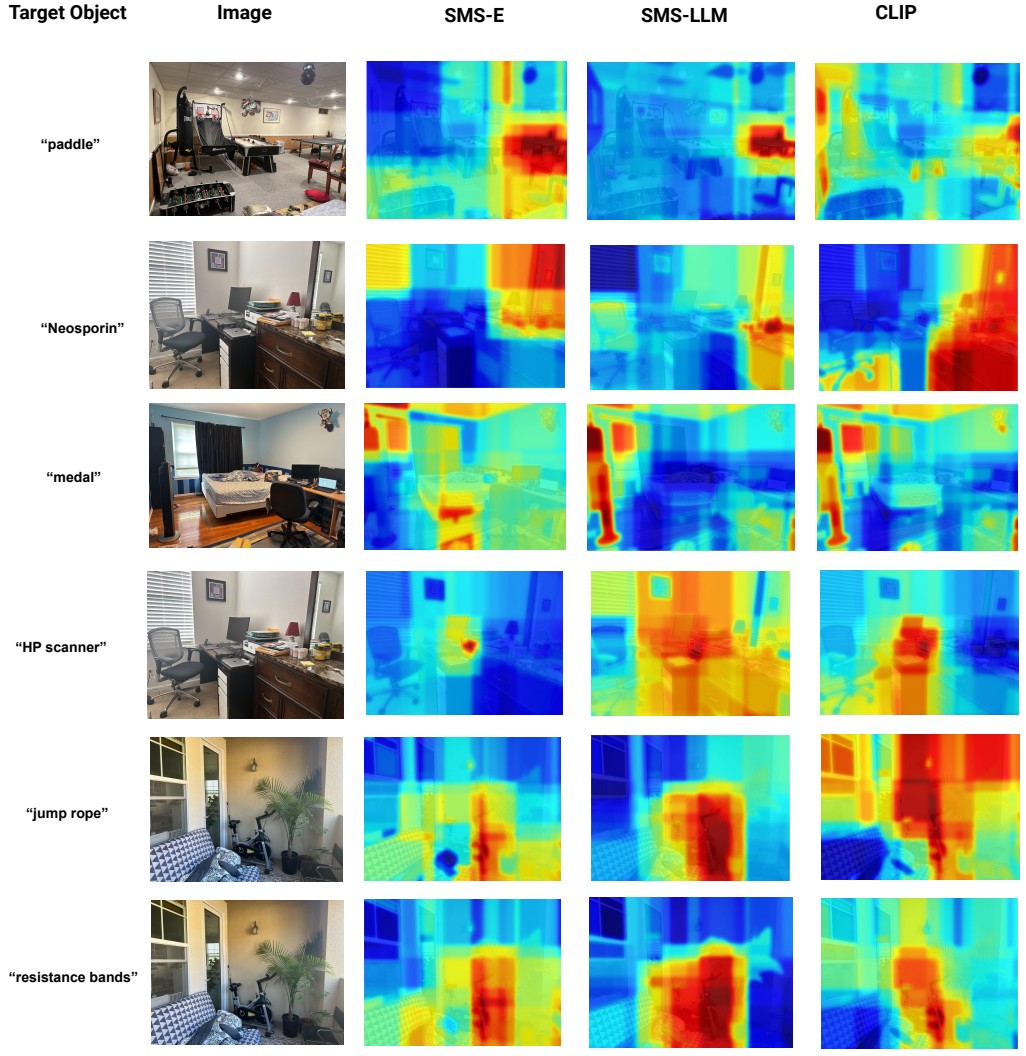

Figure 18: Example set 2 of the semantic distributions generated by different methods for houses.

