# OpenReview forum: "Semantic Mechanical Search with Large Vision and Language Models"
_robot-learning.org/CoRL/2023/Conference — CoRL 2023 Poster_

### Official Review · Reviewer_D6LF · 2023-07-18

**Confidence:** 4
**Originality:** Good
**Technical Quality:** Excellent
**Clarity Of Presentation:** Very Good
**Impact:** 3

**Recommendation:**

Strong Accept: I recommend accepting the paper and will argue for my recommendation even if other reviewers hold a different opinion.

**Review:**

This paper is well written and presents a creative and effective solution to a common problem in household robotics. The experiments are also impressive, showcasing the possibility of pretrained VLMs and LLMs to aid in increasingly sophisticated robot tasks. There is ultimately not too much to say about this work: it presents a well-scoped solution to a common problem and contains an impressive amount of experiments in both simulation and in the real world.

As such, most of my comments are mostly optional suggestions for improved readability and clarity.

The paper's biggest weakness is its density, something difficult to remedy without significant changes. As written, the paper favors building intuition via prose to including additional visuals to aid in understanding the low-level methodology and results. For example, Figure 1 is clean and nice to look at, but does not do a particularly effective job at showcasing the low-level process by which the different stages are related to one another. For instance, it was unclear until reading on page 4 (two full pages after the figure appears) what how the "semantic distribution" and "affinity scores" were related to one another. Similarly, only a single results figure is provided in the main body of the text and, while helpful, it only helps to visualize the results for Sec. 5.2. Additional figures (even those in the appendix) to help better understand the results from 5.1 would be a welcome addition.

These are perhaps smaller issues, since indeed the full method is understandable in the current version of the work and the paper stands well on its own. However, some effort from the authors to favor more visuals (or at least include more such visuals in the appendix) would likely improve the impact of the work.

Scattered other comments:
- I don't believe it is mentioned which LLM is used for these experiments. I understand these days that naming which LLM is being used can often de-anonymize authors, so an acknowledgment by the authors that this information will be included in the final version will be sufficient here.
- [typo?] The title of Fig. 2 is "Object navigation experiments in real". Perhaps this should end with "real world environments" or similar.
- The limitations section should perhaps include a comment about the quality of the individual tools upon which the proposed method relies. For example, Sec. 5.2 mentions that performance is limited by the quality of labels from BLIP-2. Reiterating this point (and similar) in limitations if there is space would be welcome, but is not necessary.


**Quality Of The Limitations Section:**

Limitations are addressed clearly

**Questions For Rebuttal:**

None. The paper is clear and easy to follow and the results well-motivated and impressive. I am happy to work with the authors during the rebuttal phase if they have thoughts about adding additional visuals and the trade offs space limitations would make necessary.

**Robotics Focus:**

Sufficient demonstration on hardware

**Summary Of Paper:**

This paper presents Open-World Semantic Mechanical Search (OWSMS), a large language model driven approach to mechanical search, in which locating and retrieving a target object may require navigation and or manipulation. CLIP approaches often fail since the objects being searched for are often dissimilar to the objects around them. The proposed approach queries a large language model for semantic information about the relative location of observed objects to the target object and then uses this "semantic distribution" to inform where to look. In image space, segmented patches from Segment Anything are used as inputs to generate the semantic distribution, which then provides affinity scores for each patch, each describing how likely the object in image space is likely to be most near to the object of interest. The affinity scores are then used for downstream tasks including shelf searching and object-goal navigation, showing improvements in both simulated and real-world environments.


**Summary Of Recommendation:**

This paper is well written and presents a creative and effective solution to a common problem in household robotics. The experiments are also impressive, showcasing the possibility of pretrained VLMs and LLMs to aid in increasingly sophisticated robot tasks. Additional visuals (and in particular a rework of Fig. 1) would greatly help readability and understanding of the impact of results.

---

### Official Review · Reviewer_5M8F · 2023-07-19

**Confidence:** 5
**Originality:** Good
**Technical Quality:** Very Good
**Clarity Of Presentation:** Very Good
**Impact:** 3

**Recommendation:**

Weak Accept: I recommend accepting the paper, but will not argue for my recommendation if the majority of other reviewers have a different opinion.

**Review:**

## Strengths
 - Mechanical search system was demonstrated on real robot
 - Large number of simulation-based evaluation episodes with standard errors for mechanical search.
 - Interesting Google Taxonomy based approach for semantic arrangement of simulation scenes. This has potential to be scalable approach to generate a large-scale simulation evaluation suite.

## Weaknesses

### Incomplete, Unsystematic Evaluation

The proposed solution is a common zero-shot recipe (e.g., [4,5]) of fusing VLMs and LLMs into a plug-and-play scene representation (e.g. [3,6]) for downstream low-level manipulation.
Since this zero-shot recipe is not novel, demonstrating that it improves over prior work without VLMs and LLMs is not a contribution.
However, applying the recipe to different robotic zero-shot tasks is still educational and informative because it establishes empirical rankings between different VLMs and how to use them for each task, usually with high statistical confidence (using lots of episodes, because pre-trained models are used zero-shot).
Further, it allows *systematic* variation in open-world properties (uncommon objects, appearance descriptions, etc. in [3], unseen objects, vocabulary, domains in [2]).
CoW [3] extensively and systematically evaluated 3 VLMs, multiple different grounding approaches with CLIP, prompts, on two different simulation benchmarks.

From empirical findings in [3-6], comparing LAX-RAY with OWSMS (Table 2, 3) is necessary but not informative.
Ablations like OWSMS-E and OWSMS-LLM are useful, but not complete (e.g., lacks insights in prompts to justify L181-183, object detectors other than ViLD).
This means the proposed approach's system vision pipeline of ViLD + OCT => SAM => BLIP-2 => CLIP is a large, unablated pipeline, making its design difficult to learn from for readers.

Further, their evaluation for unconstrained environments does not include mechanical search and only object navigation, has a small sample size (10 scenes), is not systematic (does not vary different open-world property separately), and is not reproducible (real environment not accessible for follow-up work, robot was teleoperated).

I would like to see empirical results using a simulation benchmark that is reproducible that systematically (different system level design, VLM prompt, and VLM model choices, and evaluate the same design choices for all settings, not just for unconstrained environments) ablates the proposed system.

### Lack of Visualizations

I would like to see more visualizations of the real world object navigation example scenes, and the complete list of objects in the ten scenes.
Without this, it becomes difficult to interpret the paper's custom unconstrained evaluation benchmark's quantitive results.

### Open-Vocabulary, not Open-World

From [9]'s visualizations, FOSS only contains abstract extruded cylinders and cuboids, devoid of visual-semantic meaning. This means visual distribution shifts -- a core part of open-world generalization -- is not tested.



## Areas for Improvement

 - **Evaluation**:
   - Make evaluation systematic and reproducible.
   - If the authors wish to claim a plug-and-play representation for open-world search (encompass both open-world mechanical search and object navigation) then compare approach on CoW benchmark, to allow for reproducible open-world evaluation.
   - If the authors wish to claim a plug-and-play representation for only open-world mechanical search, then provide open-world (i.e., unconstrained) evaluations for mechanical search.
 - Add more real-world unconstrained visualizations
 - Add complete list of real-world 5 real world objects from all 10 scenes in Supplementary Materials
 - **Second contribution**. "An empirically evaluated pipeline for generating object labels in open-world scenes without access to any object lists" was not ablated. For instance, using all class names in Google Taxonomy's relevant class (e.g., OWL-VIT [8] uses all category names in LVIS) is a good baseline.
 - **Fair pre-trained baseline comparisons**: large pre-trained models are useful but noisy (as mentioned in L172), and using averaging over cropping and flipping augmentations is a well explored technique. For instance, [2] found that cropping + jittering leads to better CLIP relevancy maps. The baselines in open-world evaluation should also use cropping augmentations for CLIP and OWL-VIT baselines.
 - **Writing**:
   - "For unconstrained environments, results suggest that our semantic distributions outperform those generated by CLIP-based methods" L20-22: This was neither in open-world mechanical search nor open object navigation. Meanwhile, open-world open navigation contains only one non-reproducible example.
   - LLM only approaches are mentioned in related works (L122-127), with their disadvantages being that they require non-language information and domain-specific policies. However, LAX-RAY, the "speciallized planning and manipulation [policy]" (L130) that OMSMS was designed to work with, is a domain-specific policy which uses non-language information.
   - L129 "distills" typically implies some learnable parameter. Consider changing to another word, such as "uses".
   - What is "Spatial NN" (Table 2)? NN was not defined anywhere.


## Citations

Side note: [1-2, 5] are relevant missing citations. [3-4,6-8] have been cited in the work.

 - [1] [Generic Attention-model Explainability for Interpreting Bi-Modal and Encoder-Decoder Transformers](https://hila-chefer.github.io/publication/generic-attention-model-explainability/)
 - [2] [Semantic Abstraction: Open-World 3D Scene Understanding from 2D Vision-Language Models](https://semantic-abstraction.cs.columbia.edu/)
 - [3] [CoWs on Pasture: Baselines and Benchmarks for Language-Driven Zero-Shot Object Navigation](https://cow.cs.columbia.edu/)
 - [4] [Do As I Can, Not As I Say: Grounding Language in Robotic Affordances](https://say-can.github.io/)
 - [5] [Grounded Decoding: Guiding Text Generation with Grounded Models for Robot Control](https://grounded-decoding.github.io/)
 - [6] [Open-vocabulary Queryable Scene Representations for Real World Planning](https://nlmap-saycan.github.io/)
 - [7] [LM-Nav: Robotic Navigation with Large Pre-Trained Models of Language, Vision, and Action](https://sites.google.com/view/lmnav)
 - [8] [Simple Open-Vocabulary Object Detection with Vision Transformers](https://arxiv.org/pdf/2205.06230.pdf)
 - [9] [Mechanical search on shelves using lateral access x-ray](https://research.google/pubs/pub50619/)

**Quality Of The Limitations Section:**

Limitations are addressed clearly

**Questions For Rebuttal:**

### LLM completion probability design

This is more of a minor observation than a question, but it could be discussed in the limitations on how completion probabilities is only a proxy for semantic distance.

The authors described on L183 "The affinity score for the target object with each label is the completion probability for the tokens that represent the target object".
Since completion probability for the label is proportional to exponential of sum of token log probs, longer object names will likely result in lower logprob.

For instance, using `text-davinci-003`  with probabilities enabled on the [OpenAI playground](https://platform.openai.com/playground), I used the approach's prompt to find a `toothbrush`:
```
I see the following in a room: {obj}. This is likely to be the closest object to toothbrush.
```

reliably gave the following log probabilities for `{obj}`:

- `colgate toothpaste tube`: -33.71 logprob, 5 tokens.
- `toothpaste tube`: -26.04 logprob, 3 tokens.
- `paste`: -21.23 logprob, 1 token.
- `toothpaste`: -19.83 logprob, 2 tokens.
- `tooth`: -16.87 logprob, 1 token.
- `cat`: -15.55 logprob, 1 token.

Which means the approach will prioritize searching behind the cat first.

**Robotics Focus:**

Sufficient demonstration on hardware

**Summary Of Paper:**

The paper proposed the task of open-world mechanical search, which extends mechanical search to the open-world (unseen vocabulary, unseen objects) evaluation setting.
Along with the task, the paper introduced an approach for combining off-the-shelf pre-trained VLMs, LLMs, object detectors with a prior works' mechanical search solution.
Using LLM completion logprobs as a proxy for semantic distance (in the taxonomy tree) between detected objects, the approach combines these semantic distances into one semantic map.
The manipulation policy uses a map that combines the aforementioned semantic map (1D for shelf mechanical search, 2D for object navigation) and a prior work's spatial occupancy distribution map to decided which object to manipulate.
This approach improves mechanical search success rates marginally and action-efficiency significantly on average across the pharmacy, kitchen, and office domains in simulation, as well as in real world.

**Summary Of Recommendation:**

My biggest concern is about the evaluation, which I believe can be more systematic, complete, and reproducible. As with many other open-world papers leveraging large pre-trained models, the approach lacking novelty is not a basis of rejection as long as the proposed system design is sufficiently ablated and insights about the system's design are discussed such that others can learn from and improve upon the results. With the paper's current evaluation, my recommendation is a strong reject.

**Update**
The biggest change the authors made in my opinion include a change in scope from open-world to closed-world. This reduced the systematic requirements for evaluation, my biggest concern. The authors have ran a lot of experiments which ablated various parts of the proposed pipelines in closed-world settings, and provided an un-ablated result on an open-world object navigation search task. Although the approach is still not novel, the experiments are quite thorough and well-executed.

---

### Official Review · Reviewer_jFnz · 2023-07-19

**Confidence:** 4
**Originality:** Good
**Technical Quality:** Good
**Clarity Of Presentation:** Very Good
**Impact:** 3

**Recommendation:**

Weak Reject: I recommend rejecting the paper, but will not argue for my recommendation if the majority of other reviewers have a different opinion.

**Review:**

Following up on the summary above:

When the objects are known a priori, OWSMS identifies their location in the environment using an existing object detection model (ViLD) and OCR method. Without an object list, OWSMS identifies object crops in RGB and generates their captions using BLIP-2. Given the location and label of objects in the environment, the paper proposes two approaches to generating the target object distribution $P(\textrm{target} \vert \text{label})P(\text{label})$ (I recognize that this expression is an over-simplification): OWSMS-LLM uses an LLM to approximate the likelihood of a candidate location of the target object based on the nearest labeled object $P(\textrm{target} \vert \text{label})P(\text{label})$; OWSMS-E estimates $P(\textrm{target} \vert \text{label})P(\text{label})$ using the dot product between textual embeddings of the target and label object. When the objects (labels) are unknown, $P(\textrm{label})$ is determined by taking the dot product between the image crops and the generated labels embedding, otherwise $P(\textrm{label})=1$ (i.e., when the objects are known a priori, the detections are treated as truth). The resulting semantic distribution is combined with a spatial distribution (from LAX-RAY) to get the final distribution over the target object's location. This distribution is used by a downstream policy (LAX-RAY in the context of constrained environments and a heuristic in unconstrained environments) to then search for the object.

The paper evaluates the OWSMS variants in simulated and real-world domains that involve searching for an object on a shelf (constrained environment). The paper presents additional results that involve searching a collection of images taken in an open environment (unconstrained environment). In both cases, paper shows that OWSMS outperforms CLIP and other baselines in terms of the accuracy of the resulting semantic distribution and the number of actions required to find the target object.



STRENGTHS
+ The problem of "mechanical search" is important and the idea of using LLMs to facilitate this search, notably to provide semantic information that can be used to hypothesize the location of a target object relative to a known object is sensible and interesting.
+ The experimental results demonstrate that the manner by which the method incorporates information from the LLM and the word embeddings results in semantic distributions that are more accurate and that, at least in conjunction with spatial information, improve the performance of search.
+ The paper is largely well written and easy to follow.

WEAKESSES
- The paper is missing a **large** body of work on language understanding in robotics [1-20]. The list of references in the paper is extensive, but with a significant bias towards papers from the last few years, which makes it difficult to put this work in the context. Particularly relevant are methods that similarly consider searching for a named object in a partially known environment, some using learned models of object-object interaction [3, 4, 6, 12, 1, 14] to similarly maintain distributions over the location of the target (and other unobserved objects). I don't expect the authors to cite all of these papers, but do feel that it's important to reference relevant existing work, even if it is more than a few years old :-)
- Similarly, the paper omits relevant work in object (active visual) search [21-24], which includes methods that exploit learned semantics and object-object cooccurrence facilitate search. That said, it is worth noting that the authors have cited papers that are more than just very recent papers.
- The scale of the environments considered during evaluation are particularly small in both size (e.g., 0.8m x 0.35m x 0.57m for the shelf experiments) and the number of objects.
- The OWSMS-E variant is sensible, but also rather obvious, namely using textual embeddings as a measure of co-occurence. Thus, the significance of this contribution is limited.
- It's not obvious that OWSMS-LLM is demonstrably better than OWSMS-E, the novelty of which is limited, in terms of search. In some cases, OWSMS-LLM outperforms OWSMS-E, in others OWSMS-E performs comparably or even better.
- Indeed, the title and algorithm name emphasize "open-world" however the arguably straightforward embedding-based version performs better than the LLM variant in the open-world domain, while the latter performs best in the closed-world (known object list) environment.
- The paper ablates access to an object list only with regards to the resulting distribution relative to a reference distribution based on the Google Taxonomy (the relevance of which as ground-truth is unclear). However, it does not evaluate the effect of task performance (e.g., success rate or number of actions) when the object list is not known. Meanwhile, it is not obvious that the results in Table 4 contribute much. As the paper notes, one would expect the distribution to be better when the list of objects is known a priori.
- Related, my understanding is that the LAX-RAY baseline does not require the other objects to be known a priori (i.e., no object list). If this is the case, then the results in Tables 2 and 3 don't provide an apples-to-apples comparison.
- It isn't obvious how the high-level problem considered here differs from active visual search. Assuming that the differences are minimal, why does the problem need to be rebranded as "mechanical search"?



MINOR
* Line 208: "the manipulating" --> "manipulating"
* Table 2: The OWSMS-LLM numbers for Kitchen should not be bold as the differences relative to OWSMS-E are not statistically significant.

REFERENCES

[1] P. Anderson, Q. Wu, D. Teney, J. Bruce, M. Johnson, N. Sünderhauf, I. D. Reid, S. Gould, and A. van den Hengel. Vision-and-language navigation: Interpreting visually-grounded navigation instructions in real environments. In Proceedings of the IEEE/CVF Conference on Computer Vision and Pattern Recognition (CVPR), 2017.

[2] D. L. Chen and R. J. Mooney. Learning to interpret natural language navigation instructions from observations. In Proceedings of the National Conference on Artificial Intelligence (AAAI), 2011.

[3] F. Duvallet, T. Kollar, and A. Stentz. Imitation learning for natural language direction following through unknown environments. In Proceedings of the IEEE International Conf. on Robotics and Automation (ICRA), 2013.

[4] F. Duvallet, M. R. Walter, T. Howard, S. Hemachandra, J. Oh, S. Teller, N. Roy, and A. Stentz. Inferring Maps and Behaviors from Natural Language Instructions. In Proceedings of the International Symposium on Experimental Robotics (ISER) June 2014

[5] D. Fried, R. Hu, V. Cirik, A. Rohrbach, J. Andreas, L.-P. Morency, T. Berg-Kirkpatrick, K. Saenko, D. Klein, and T. Darrell. Speaker-follower models for vision-and-language naviga- tion. In Advances in Neural Information Processing Systems (NeurIPS), Dec. 2018.

[6] S. Hemachandra, F. Duvallet, T. M. Howard, N. Roy, A. Stentz, and M. R. Walter. Learning models for following natural language directions in unknown environments. In Proceedings of the IEEE International Conference on Robotics and Automation (ICRA), 2015.

[7] T. M. Howard, S. Tellex, and N. Roy. A natural language planner interface for mobile manipula- tors. In Proceedings of the IEEE International Conference on Robotics and Automation (ICRA), 2014.

[8] T. Kollar, S. Tellex, D. Roy, and N. Roy. Toward understanding natural language directions. In Proceedings of the ACM/IEEE International Conference on Human-Robot Interaction (HRI), 2010.

[9] M.MacMahon, B. Stankiewicz, and B. Kuipers. Walk the talk: Connecting language, knowledge, and action in route instructions. In Proceedings of the National Conference on Artificial Intelligence (AAAI), 2006.

[10] C. Matuszek, D. Fox, and K. Koscher. Following directions using statistical machine translation. In Proceedings of the ACM/IEEE International Conference on Human-Robot Interaction (HRI), 2010.

[11] C. Matuszek, E. Herbst, L. Zettlemoyer, and D. Fox. Learning to parse natural language commands to a robot control system. In Proceedings of the International Symposium on Experimental Robotics (ISER), 2012.

[12] H. Mei, M. Bansal, and M. Walter. Listen, attend, and walk: Neural mapping of navigational instructions to action sequences. In Proceedings of the National Conference on Artificial Intelligence (AAAI), 2016.

[13] D. K. Misra, J. Sung, K. Lee, and A. Saxena. Tell me Dave: Context-sensitive grounding of natural language to manipulation instructions. International Journal of Robotics Research, 35 (1-3):281–300, January 2016.

[14] S. Patki, E. Fahnestock, T. M. Howard, and M. R. Walter. Language-guided Semantic Mapping and Mobile Manipulation in Partially Observable Environments. In Proceedings of the Conference on Robot Learning (CoRL) October 2019

[15] R. Paul, J. Arkin, D. Aksaray, N. Roy, and T. M. Howard. Efficient grounding of abstract spatial concepts for natural language interaction with robot platforms. International Journal of Robotics Research, 37(10):1269–1299, June 2018.

[16] M. Shridhar and D. Hsu. Interactive visual grounding of referring expressions for human-robot interaction. In Proceedings of Robotics: Science and Systems (RSS), 2018.

[17] S. Tellex, T. Kollar, S. Dickerson, M. R. Walter, A. G. Banerjee, S. Teller, and N. Roy. Un- derstanding natural language commands for robotic navigation and mobile manipulation. In Proceedings of the National Conference on Artificial Intelligence (AAAI), 2011.

[18] J. Thomason, S. Zhang, R. J. Mooney, and P. Stone. Learning to interpret natural language commands through human-robot dialog. In Proceedings of the International Joint Conference on Artificial Intelligence (IJCAI), 2015.

[19] J. Thomason, J. Sinapov, M. Svetlik, P. Stone, and R. J. Mooney. Learning multi-modal grounded linguistic semantics by playing “I spy”. In Proceedings of the International Joint Conference on Artificial Intelligence (IJCAI), 2016.

[20] J. Thomason, J. Sinapov, R. J. Mooney, and P. Stone. Guiding exploratory behaviors for multi-modal grounding of linguistic descriptions. In Proceedings of the National Conference on Artificial Intelligence (AAAI), 2018.

[21] A. Aydemir, A. Pronobis, M Göbelbecker, and P. Jensfelt. Active visual object search in unknown environments using uncertain semantics" IEEE Transactions on Robotics 29, no. 4 (2013): 986-1002.

[22] J. K. Tsotsos.On the relative complexity of active vs. passive visual search. International Journal of Computer Vision 7, no. 2 (1992): 127-141.

[23] L. E. Wixson, and D. H. Ballard. Using intermediate objects to improve the efficiency of visual search. International Journal of Computer Vision 12, no. 2-3 (1994): 209-230.

[24] S. Ekvall, D. Kragic, and O, Jensfelt. Object detection and mapping for service robot tasks. Robotica 25, no. 2 (2007): 175-187.



UPDATE AFTER AUTHOR RESPONSE AND DISCUSSION

The authors clearly made an effort to respond to the questions and concerns that I raised in my initial review and the follow-up discussion.

The authors addressed some of the issues that I initially identified. However, I still feel that the paper should be evaluated as a systems paper that describes the integration of existing methods to perform semantic search for occluded objects. As I mentioned in my reply to the authors, it is my view that a systems paper should either provide novel insights into the problem of mechanical (active visual search), e.g., through a compelling experimental evaluation, or show how using existing methods requires combining them in novel ways. I'm just not sure that the paper does this. The paper lacks comparisons to relevant active visual search methods and instead compares to OWL-ViT and a CLIP baseline that, to the best of my knowledge, are not designed to deal with occlusions. Are the

I note that regarding the discussion of labeling the method as "mechanical search " vs. "active visual search" the authors point to the definition of active visual search (AVS) in Aydemir et al. [1] and suggest that it differentiates from mechanical search in that the latter must deal with occlusions. However Aydemir et al. also discuss the need to mitigate occlusions: "Some of the advantages of an active strategy discussed are robustness to occlusions, possible increase of resolution, and use of motion to disambiguate vision-related aspects of the world such as varying illumination conditions." Later, in differentiating their work from a specific POMDP-based search strategy [2], they state that "This [visibility assumptions that Hollinger et al. [2] make] is a big simplification of detecting an object with a camera since the task of finding an object in a place as big as a room involves many difficulties such as calculating a good viewing position, dealing with occlusions, and detecting objects that appear small in the image." I fail to see a fundamental distinction between searching for an occluded object on a shelf using a robot arm and searching for an occluded object in a multi-room environment. Again, this goes beyond the simple question of whether "mechanical search" vs "active visual search" is appropriate and instead relates to the importance of putting this work in context.

Minor: I thank the authors for looking at the myriad of papers that I listed in my initial review. Admittedly, I may have gone a bit overboard with such a long list and It was not my intention that the authors cite most of these papers, but rather the ones that are particularly relevant with regards to using information extracted from textual descriptions to find an object that is not visible to the robot and the location of which is unknown. This goes beyond instances of natural language understanding as cited in the revised paper.

[1] Aydemir, Alper, Andrzej Pronobis, Moritz Göbelbecker, and Patric Jensfelt. "Active visual object search in unknown environments using uncertain semantics." IEEE Transactions on Robotics 29, no. 4 (2013): 986-1002

[2] G. Hollinger, D. Ferguson, and S. Srinivasa, S. Singh, “Combining search and action for mobile robots,” in Proc. IEEE Int. Conf. Robot. Autom., May 2009, pp. 800–805.

**Quality Of The Limitations Section:**

Additional details required

**Questions For Rebuttal:**

1) Section 3: Can the authors clarify what they mean by the assumption that "the scenes in the environment are semantically organized, meaning that the starting state of the environment is sampled proportionally to their approximate likelihood of occurrence in the real world"?

2) Section 3: Does the method assume that the geometry of the objects is known exactly? If not, how are the geometries inferred from observations?

3) Section 4.2: How many candidate completions from the LLM does OWSMS-LLM consider?

4) Is it correct that the comparisons to the LAX-RAY baseline provide an ablation of the spatial distribution (i.e., running a version of OWSMS w/o the semantic distribution)?

5) Can the authors justify the use of the Google Taxonomy-based distribution as ground-truth?

6) Section 5.2: How is the experimental setup chosen, namely the number and placement of objects and the choice of target objects? One questions whether the scenarios may have been cherry picked.

**Robotics Focus:**

Sufficient demonstration on hardware

**Summary Of Paper:**

The paper considers the problem of active visual search in which a robot is tasked with searching for a named object that may be occluded or is otherwise not immediately visible to the robot and for which its pose is unknown. The paper specifically focuses on learning an occupancy map that models the distribution over the object's location that can be used by a downstream method to plan and execute the search. To that end, the paper proposes Open-World Semantic Mechanical Search (OWSMS), a framework that builds and maintains a pixel-level occupancy distribution over the target object's location based on its semantic similarities to visible objects in the scene, where semantic similarity is modeled in terms of object-object co-occurrence determined either using an LLM or textual embeddings.

**Summary Of Recommendation:**

The idea of using language models to facilitate active search is interesting and sensible. This work has potential, but the lack of qualitative comparisons to missing related work together with the inconclusive nature of the experimental results make it difficult to judge the significance of the contributions.


SEE ABOVE FOR UPDATES AFTER AUTHOR RESPONSE/DISCUSSION

---

### Official Review · Reviewer_kaAs · 2023-07-20

**Confidence:** 4
**Originality:** Good
**Technical Quality:** Good
**Clarity Of Presentation:** Good
**Impact:** 3

**Recommendation:**

Weak Accept: I recommend accepting the paper, but will not argue for my recommendation if the majority of other reviewers have a different opinion.

**Review:**

Strengths:
- Generally, the structure of the paper is clear and it is reasonable to take advantage of the proposed OWSMS as a plug-in semantic module instead of an end-to-end planners. It is a good application of LLMs in robotic tasks.

Weaknesses:
- It is said that the scenarios used in this task are pharmacy shelves and kitchen cabinets, while as seen from the paper and the accompanied video, the scenarios are actually not very challenging and somewhat artificial. In these circumstances, I have some doubt about how important is the role of the semantic occupancy distribution here. And the paper did not present a specific affinity matrix. I think it would be helpful for me to evaluate the role of the semantic occupancy distribution if a detailed affinity matrix with objects detected from the scene for a certain scenario is presented.

- In the experiments, the proposed methods are compared with the LAX-RAY method and some improvement is achieved, while the improvement is actually not very obvious. I think it would be helpful to have a Random baseline in the experiments, as it would reflect how difficult the task is to some extend. And it can also be regarded as an ablation study which evaluates the  impact of the semantic occupancy distribution in the framework.

- Some minor problems such how the state $s_t$ encodes the full geometries, poses, and names of the objects in the scene, some visualization of the scenarios and  semantic occupancy distribution.



**Quality Of The Limitations Section:**

Additional details required

**Questions For Rebuttal:**

As stated in the Weaknesses above.

**Robotics Focus:**

Sufficient demonstration on hardware

**Summary Of Paper:**

In this paper, an Open-World Semantic Mechanical Search (OWSMS) framework which leverages the LLMs to generate a semantic occupancy distribution for the target object. And the physical experiments to search object on shelves are conducted.

**Summary Of Recommendation:**

This paper provide a reasonable direction to leverage LLMs to practical robotic tasks, while the scenarios are somewhat simple and the impact of the semantic occupancy distribution should be further explained.

---

### Author Response · Authors · 2023-08-11
**General Summary of the Rebuttal**

***Please see the updated manuscript and appendix attached to this comment.***

We thank all reviewers for their helpful comments and suggestions. We are glad that most of the reviewers thought our algorithm and hardware demonstrations interesting and compelling, and, as one of the reviewers points out, “The problem of mechanical search is important and the idea of using LLMs to facilitate this search, notably to provide semantic information that can be used to hypothesize the location of a target object relative to a known object is sensible and interesting”

We revised the manuscript and the appendix to address the reviewer's suggestions. All changes are highlighted in blue in the revised manuscript. Specifically, we made the following changes:

1. We add experiments comparing different LLMs for affinity matrix generation for constrained environments in appendix Table 8 as suggested by Reviewer kaAs.
2. We increase the number of scenes used in unconstrained environment experiments dataset in Sec. 5.2 . All the scenes and target object list are included in the appendix Sec. F.
3. We add augmentations including jittering and flipping to baselines as suggested by Reviewer 5M8F and update the results in Table 5 with the new dataset.
4. We add ablation studies on different modules of our framework and study the impact of each module in Table 6 as suggested by Reviewer 5M8F.
5. We implement OWSMS in the CoW benchmark as suggested by Reviewer 5M8F and report the results in appendix Table 10.
6. We add more visualizations in the appendix as suggested by Reviewers kaAs and D6LF to clarify OWSMS pipeline in shelf environments (Figs 4, 7) and to show examples of shelf environments layout in simulation and in real (Figs 5, 6) as suggested by Reviewer kaAs.

We look forward to continued discussion with the reviewers about our updated manuscript.

---

### Decision · Program_Chairs · 2023-08-30

**Decision:**

Accept (Poster)

**Comment:**

The paper introduces a new method for semantic mechanical search using large vision and language models.

The paper causes heat discussion during the rebuttal period and the reviewer discussion period.
In the end, majority of  the reviewers agree on accepting the paper. However, there is still concerns on the novelty of the proposed method.
Therefore, I recommend accepting the paper as a poster presentation.